# Estimation of Arctic Land-Fast Ice Cover based on Dual-Polarized SENTINEL-1 SAR Imagery

Juha Karvonen[1]

[1]Finnish Meteorological Institute, PB 503, FI-00101, Helsinki, Finland

**Correspondence:** Juha Karvonen (juha.karvonen@fmi.fi)

**Abstract.** Here a method for estimating the land-fast ice (LFI) extent from dual-polarized SENTINEL-1 SAR mosaics of an Arctic study area over the Kara and Barents Seas is presented. The method is based on temporal cross-correlation between adjacent daily SAR mosaics. The results are compared to the LFI of the Russian Arctic-Antarctic Research Institute (AARI) ice charts. Two versions of the method were studied: in the first version (FMI-A) the overall performance was optimized and

in the second version (FMI-B) the target was a low LFI misdetection rate. FMI-A detected some over 73 % of the AARI ice chart LFI, and FMI-B a little over 50 % of the AARI ice chart LFI. During the winter months the detection rates were higher than during the melt-down season for both the studied algorithm versions. An LFI time series covering the time period from October 2015 to the end of August 2017 computed using the proposed methodology is provided on the FMI ftp server. The time series will be extended twice annually.

## 1  Introduction

Land-fast ice (also known as shore-fast ice, or shortly as fast ice), here denoted by LFI, is sea ice attached to the coastline, to the sea floor along shallow areas or to grounded icebergs [WMO (2015); Weeks (2010); Lepparanta (2011)]. LFI may either grow in place from the sea water or by freezing drifting ice to the shore [WMO (2015)]. LFI does not move with currents and winds. LFI zone is typically seasonal and depends on ice thickness, topography of the sea floor and islands [Lepparanta

(2011)]. On average the fast ice edge is located in the water depth of 10-25 m [Zubov (1945); Divine et al. (2004); Mahoney et al. (2007)]. However, there exists seasonal and inter-annual variability. Based on the numbers given e.g. in [Yu et al. (2014)] LFI area covers approximately 13% of the Northern Hemisphere area of sea ice cover, and thus represents an essential fraction of the Arctic sea ice. LFI zone distance from coast varies from a few meters to several hundreds of kilometers [WMO (2015)]. For practical LFI detection some criteria to detect the LFI areas need to be fixed. In [Mahoney et al. (2005)] two criteria have

been used for LFI: the ice is contiguous with land and it lacks detectable motion for approximately 20 days. In the methodology presented here a two-week period without ice motion and contiguousness with land have been used as criteria for LFI.

Long-term changes have been found in the LFI regime. The trend seems to be toward reducing LFI area [Divine et al. (2003); Yu et al. (2014)], later formation and earlier disappearance [Mahoney et al. (2014); Seluyzhenok et al. (2015)] and reduction of the LFI thickness [Polyakov et al. (2003, 2012)]. Although LFI zone only covers a relatively small fraction of

the total Arctic sea ice extent, it has particular importance for the coastal systems, e.g. by defining the location of polynyas

[Morales Maqueda et al. (2004)]. These facts make monitoring of the Arctic LFI zone important, also as a climate change indicator.

LFI detection based on different techniques and different instruments have been proposed earlier. Passive microwave (PM) data has widely been used for determining sea ice motion, e.g. in [Agnew et al. (1997); Kwok et al. (1998)], but because of their low spatial resolution (5-50 km) PM data have not much been used for fast ice detection [Fraser et al. (2011)]. Some examples of using PM data for LFI estimation exist e.g. temporal correlation median of AMSR-E imagery was used for LFI detection in [Seluyzhenok (2011)]. High resolution near-infrared imagery from Landsat I and II have been used for identification of Alaska LFI as sea ice contiguous with the coast [Barry et al. (1979); Stringer et al. (1978, 1980)]. A method for estimating LFI using cloudless specroradiometer (MODIS) data was proposed in [Fraser et al. (2011)]. The method uses a 20-day composite of MODIS imagery of eastern Antarctic coast additionally supported by AMSR-E ASI algorithm [Spreen et al. (2008)] sea ice concentration in the case of unreliable (possibly cloudy) MODIS image composites. In [Kim et al. (2015)] machine learning (Random Forest algorithm) using data from multiple instruments: AMSR-E brightness temperature, MODIS ice surface temperature (IST) and SSMI (Special sensor microwave/imager) ice velocity were applied to detect LFI. In the study ice velocity and IST proved to be the most significant factors in LFI detection.

Also different methods utilizing SAR imagery for LFI detection have been proposed. Significant advantages of using SAR imagery are the high resolution, typically from tens to a few hundreds of meters, of SAR imagery and the capability to measure in cloudy or dark (no daylight) conditions. SENTINEL-1 temporal cover in the Arctic is comparable to that of a radiometer (e.g. AMSR-2) data. In [Antonova (1997)] the areas of static ice were determined manually from consecutive SAR images (time series). In [Mahoney et al. (2004, 2005)] LFI was detected based on vector grayscale gradient fields of three subsequent SAR images. The bottom fast ice zone can be identified based on the SAR backscatter magnitude [Eicken et al. (2005); Solomon et al. (2005)] because if there is no ice-water interface the dielectric contrast at the bottom is significantly reduced. Ice drift can also be derived from multi-temporal SAR image pairs over the same area. Such SAR ice drift detection algorithms are typically based on temporal cross-correlation, i.e. cross-correlation between co-registered spatially (partly) overlapping SAR images acquired at different time instants [Fily and Rothrock (1987)], temporal phase-correlation [Thomas et al. (2008)] or optical flow [Sun (1996)]. From time series of ice drift estimates it is possible to derive the static ice areas which can then be interpreted as LFI, assuming the time series of ice drift at a certain location is long enough. Also SAR interferometry can be used for LFI detection [Mayer et al. (2011); Marbouti et al. (2017)], as the phase difference is random for drift ice and coherent for the static ice fields. However, the availability of Single-Look-Complex (SLC) SAR data required for SAR interferometry is currently restricted, and thus methods based on SAR interferometry are not yet suitable for LFI monitoring in a large spatial scale. In [Karvonen (2012)] the cumulative Baltic sea ice drift estimated from multi-temporal SAR imagery was used for locating the Baltic sea LFI by indicating the areas where no ice motion had occurred within a predefined and long-enough time period (typically around two weeks). In [Karvonen (2014)] temporal cross-correlation minimum was used to locate LFI to aid sea ice concentration estimation.

The algorithms proposed in this study are used for creating daily time series of the Kara and Barents Sea LFI extent in a high resolution (500 m) gradually complementing the existing Arctic LFI time series derivable from Arctic operational ice

charts. FMI has some sea ice products which have been run in an operational test mode during a few winters over the studied area. The aim is to include the LFI estimation in the FMI operational Arctic sea ice product portfolio. Technical details of the FMI operational Arctic sea ice ice thickness and concentration test products can be found in [Simila et al. (2016); Makynen and Karvonen (2018)]. One purpose is to utilize the LFI algorithm result first to locate the static ice fields and then apply the

FMI HIGH-resolution Thermodynamic Snow/Ice model (HIGHTSI) [Launiainen and Cheng (1998); Cheng et al. (2003)] to estimate the ice growth or melt during the static ice periods to improve the ice thickness estimates over the static ice areas. In the static parts of the LFI zone (during the long static periods) only thermodynamic ice modeling can be applied as the dynamic part can be omitted for the static ice. This will increase the reliability of ice modeling in the static ice areas, assuming that the static ice areas can reliably be located, as the ice thickness uncertainties originating from the ice dynamics will then be

excluded.

## 2 Study Area, Data Sets and Pre-Processing

### 2.1 Study area

The study area is located in the Kara and Barents Seas. The study area is shown in Fig. 1. The coordinate system (CS) used in this paper is the polar stereographic projection, with a center longitude of $55^{o}$E, reference latitude (latitude of the correct scale)

of $70^{o}$N and the WGS84 datum. The upper left (UL) and lower right (LR) coordinates in this CS are (polar stereographic CS northing and easting in meters): UL=(-700000,-1100000) and LR=(-2550000,1100000).

### 2.2 Russian Ice Charts

The Russian Arctic ice charts are provided weekly by AARI on their web page (the English version on http://www.aari.ru/odata/_d0015.php). They are provided as thematic maps and in SIGRID-3 vector format [JCOMM (2014)] in the polar stereographic projection

with the mid-longitude of $90^{o}$E. In this study the AARI ice chart thematic maps were reprojected into the polar stereographic projection used in this study and the LFI areas were extracted based on the colormap of the AARI ice charts. In this study 51 AARI ice charts of the period from November 2015 to October 2016, covering a whole year time, were used as reference LFI data for defining the algorithm parameters and to evaluate the proposed algorithms. Actually, four AARI ice charts (one ice chart of each month in the period January-April 2016) were used for defining the optimal algorithm threshold parameters,

and the remaining 47 ice charts were used for evaluation. The study area was cropped from the weekly reprojected AARI ice charts, and the cropped images were converted into binary 1-bit per pixel images in which the LFI areas, appearing as white in the AARI ice chart maps (for an example of an AARI ice chart see Fig. 6), were mapped to the pixel value of one and the rest of the image were mapped to the pixel value of zero.

## 2.3  SENTINEL-1 imagery, SAR Mosaic and their processing

All the available European Space Agency's (ESA) C-band SENTINEL-1 dual-polarized Extra Wide (EW) swath mode level 1 Ground Range Detected Medium resolution (GRDM) data with the HH/HV polarization channels over the study area during the study period (October 2015 - August 2017) were used in this study. The SENTINEL-1 SAR data are publicly available through the Copernicus Science Hub (https://scihub.copernicus.eu/). The imagery were preprocessed by applying an incidence angle correction to the HH channel and a combined incidence angle and noise floor correction to the HV channel, for details of this process, see [Karvonen (2017)]. After incidence angle and noise floor corrections the image data were geo-rectified into the polar stereographic projection specified in Section 2.1. After geo-rectification the imagery were still down-sampled to 500 m resolution, and finally the daily mosaics were constructed by overlaying the newer images over the older ones such that for each mosaic grid cell (pixel) the newest SAR data prior to the mosaic time label, which was defined to be 12:00 UTC daily here, was assigned to it. The mosaics were cumulative, meaning that the newer imagery was always overlaid over the previous mosaic and the mosaic was initialized only in the beginning of the mosaicking (in this case in the beginning October 2015). In practice the data at a given grid cell location were not older than three days compared to the mosaic time label, as SENTINEL-1 temporal cover over the European Arctic is so good. Separate mosaics for HH and HV channels were constructed. A land mask based on the GSHHG coastline data set [Wessel and Smith (1996)] was applied to the mosaics to exclude land areas from LFI computation. As an example of SAR mosaics the mosaics for HH and HV channels of March 8, 2016 with the land masking are shown in Fig. 2. Dual-polarized EW mode SENTINEL-1 data are systematically acquired over the European Arctic and Greenland waters by ESA, but over the other Arctic areas a single-polarization (HH) mode is used. Near the upper right corner of the study area there were no dual-polarized SENTINEL-1 EW mode data available, because this area belongs to the single-polarization HH mode acquisition area defined by ESA. This can also be seen as the black area in Fig. 2.

## 3  Methodology for estimation of land fast ice areas

The proposed LFI estimation process for a single SAR channel is presented in Fig. 3. Similar processing is performed for both the SAR channels (HH and HV) and the results are combined after computing the channel-wise LFI estimates. In the first phase the SAR mosaics are generated as described in Section 2. These mosaics (red boxes in Fig. 3) of a two week period are the inputs to the cross-correlation computation phase (indicated by yellow color in Fig. 3).

To increase the computation performance and to exclude areas where LFI does not appear a mask indicating the potential LFI zone as the areas of 100 km or less from the coastline (including islands) was produced. The mask was produced iteratively starting from the coastline, indicated by the land mask. The distance to the (nearest) coast was iteratively increased by 500 m (pixel size) for the vertical and horizontal neighbor pixels and by $\sqrt{2}{\times}500$ m for the diagonal neighbor pixels of the pixels with a distance from the coast already assigned to them. This was iterated until there were no more distances less than 100 km (corresponding to 200 grid pixels) from coast within the study area grid. The use of the proposed mask did not have any effect on the LFI detection and it was used here just to fasten the processing. The execution time for a single day LFI estimation was not very long (some minutes), but for longer LFI time series the difference of execution times with the mask and without

the mask was significant, and therefore the mask has systematically been applied in this study. The masking here applied this simple approach, however, in some other areas a more sophisticated mask taking into account the bathymetry e.g. using a given distance from the depth of 25 m, might be more useful. The mask applied is shown in Fig. 4. Unfortunately FMI did not have very accurate bathymetry data over the study area available, and thus only this simple mask was applied here. With a more accurate mask possibly more time could be saved in computation of LFI time series. This masking step is indicated by green color in Fig. 3.

The temporal cross-correlation, denoted by $C_T$, between SAR mosaics of two adjacent days was computed as

$$C_T(r,c,t) = \frac{1}{\sigma(r,c,t)\sigma(r,c,t-1)} \sum_{i,j \in W} (M(r+i,c+j,t) - \mu(r,c,t))(M(r+i,c+j,t-1) - \mu(r,c,t-1)). \tag{1}$$

The indices r and c refer to the pixel location (row and column coordinates), t refers to the day (t-1 refers to the previous day of the day t), $C_T$ is computed within a round-shaped window $W$ with a radius $R$ around the pixel at the location (r,c). In this study the value $R = 3$ was used. M(r,c,t) refers to a mosaic pixel value at the location (r,c) on day t. The means $\mu(r,c,t)$ and standard deviations $\sigma(r,c,t)$ are computed over $W$. The mosaics of the two adjacent days in the computation of $C_T$ were also always mosaics of the same polarization, either HH or HV. The cross-correlation computation is indicated by yellow color in Fig. 3. To reduce computation pixel-wise $C_T$ between two adjacent days' mosaics was also computed only over the areas defined by the distance (from coast) mask, i,e, white areas in Fig. 4.

After computing the $C_T$ grids the temporal 14 day average (blue box in Fig. 3) of the daily $C_T$ grids are computed. The areas where $C_T$ is close to one (higher than 0.95) are excluded from the average computation, as they represent areas where the mosaic has not updated since the previous day. For SAR data from two different SAR images $C_T$ is in practice always less than one, and even less than 0.95, which is used as a threshold here, because of the speckle present in all radar imagery $C_T$ is decreased even for a static target present at the same location in both of the mosaics of a mosaic pair. The initial decision whether a pixel represents LFI or doesn't is made based on thresholding of the $T_C$ average (red color in Fig. 3). The thresholds for the HH and HV SAR channels were studied by varying the threshold value and then comparing the LFI area detected by thresholding and LFI area of four AARI ice charts of the period January-April 2016, one ice chart for each month were used for defining the thresholds. The optimal thresholds were defined by minimizing the estimation error, i.e. the sum of LFI not detected by the algorithm and non-LFI classified to LFI by the algorithm when compared to the AARI ice chart LFI. The optimal thresholds yielded were $T_{HH}$=0.31 and $T_{HV}$= 0.24 for HH and HV channel mosaics, respectively. The classification error as a function of $C_T$ for both the channels can be seen in Fig. 5. These curves can be used for deriving the thresholds $T_{HH}$ and $T_{HV}$. Here we have applied a criterion minimizing the total classification error but also other criteria, depending on the objective, could be considered. At this stage a grid cell is considered as possible LFI pixel if the conditions for the channel-wise cross-correlation averages ($\overline{C}_T^{HH}$ and $\overline{C}_T^{HV}$) $\overline{C}_T^{HH} > T_{HH}$ and $\overline{C}_T^{HV} > T_{HV}$ apply and they are in the area defined by the distance mask (white areas of Fig. 4).

After applying the thresholding to $\overline{C}_T^{HH}$ and $\overline{C}_T^{HV}$ a morphological opening operation (i.e. an erosion operation followed by a dilation operation) by a disk with a radius of two pixels is applied to remove narrow elongated high $\overline{C}_T$ segments and small single patches. This operation is indicated the gray color in Fig. 3. Narrow elongated segments may appear due to the

boundaries of SAR frames over open water, where the incidence angle correction often fails because of varying wave conditions in different SAR frames in a mosaic. Small single patches may be caused by random noise or be due to possible static small targets. After applying the morphological opening operator removal of small LFI segments is applied. This filtering stage is indicated by the white box in Fig. 3. The small segment filtering performs removal of segment smaller than a given threshold

value ($T_S$). Here the value $T_s$ = 100 pixels, corresponding to an area of 25 km$^2$, has been applied. This post-filtering also efficiently reduces the number small erroneous segments due to SAR artifacts and speckle. The same procedure is performed for the HH and HV SAR channels with the only difference of applying a different threshold value ($T_{HH}$ or $T_{HV}$) depending on the SAR channel in thresholding of the temporal cross-correlation average ($\overline{C}_T^{HH}$ or $\overline{C}_T^{HV}$).

After retrieving the channel-wise LFI area estimates for the HH and HV channels the LFI estimation results of the two

SAR channels ($LFI_{HH}$ and $LFI_{HV}$) are combined by applying a logical AND operator between the channel-wise binary classification results. Finally, areas which are not connected (in the sense of 8-pixel neighborhood) to land area defined by the land mask are excluded from the LFI class. This is in practice performed by applying a recursive flood-fill algorithm [Hearn and Baker (1997)] testing the filled pixel neighbors for land (land mask pixels) while filling each contiguous LFI candidate segment produced by the earlier preliminary classification. This result is referred here as method A, shortly FMI-A.

To further reduce the erroneous non-LFI classification to the LFI category by FMI-A, an additional temporal logical AND operation applied to fourteen adjacent day FMI-A products was performed. Applying of the logical AND gives the areas where there has been LFI in each daily FMI-A classification during the two-week period. The result after the logical AND operation is referred here as method B (FMI-B).

## 4   Results

The results were first computed for a test data set over a one year a period from November 2015 to November 2016, and the results were compared with the weekly AARI ice charts. The comparison was pixel-based and it was performed between the daily LFI products and the corresponding AARI LFI of the same date (ice chart issuing date). The daily LFI products were the LFI estimates produced by FMI-A and FMI-B using the SAR mosaic of the LFI product issuing date and SAR mosaics of the preceding two week time period. Totally 47 weekly AARI ice charts were used in the comparison (the exact period was

from November 3 2015 to November 1 2016). Four weekly AARI ice charts used in defining the algorithm thresholds were excluded from the numerical comparisons. Because FMI-A also suggested a little amount of LFI during the summer, and these summer LFI areas were the same areas for both the summers included in the study, the summer LFI areas suggested by FMI-A in August 2016 were filtered out (subtracted) from all the FMI-A and FMI-B products. It was assumed that these summer LFI detections were due to inaccuracies in the land mask.

The results of the comparison of FMI-A and FMI-B LFI to AARI ice chart LFI can be seen in Table 1. In the first column are the fractions (in percents) of the LFI classified correctly when compared to AARI ice chart grids for FMI-A and FMI-B, and in the second column are the relative amounts (with respect to the AARI ice chart LFI extent) of grid points, which are not LFI according to the AARI ice charts but still classified to LFI by the proposed algorithms. In the parentheses are the standard

deviations over all the weekly cases included in the comparison. It can be seen that FMI-A is able to locate over 73% of the LFI indicated by the AARI ice charts, and in addition some over 20% of additional LFI areas were suggested by FMI-A compared to the AARI LFI area. FMI-B only detects some over half of the LFI suggested by the AARI ice charts, but very few areas outside the AARI ice chart LFI area are classified to LFI by FMI-B. An example of the LFI extent based on the cropped AARI

ice chart of March 8, 2016 in Fig. 6 and FMI-A and FMI-B LFI estimates of the same day are shown in Fig. 7. It can be seen that basically AARI ice chart LFI and FMI-A cover the same areas, but there still are some differences near the boundaries of the detected LFI area. This kind of differences occurring in most of the cases, at least partly explaining the differences in classification rates in Table 1. The total LFI extent detected by FMI-A and LFI extent given by the AARI ice charts are similar. FMI-B detects significantly less LFI than present in the AARI ice charts, but still the LFI area locations agree well, just the

LFI area being smaller in the detection results of LFI-B than in the AARI ice charts.

Also a monthly comparisons between the AARI ice chart LFI and FMI-A and FMI-B LFI were made. The results show that the FMI-A LFI estimates covered about 80% or more of the AARI LFI during the winter months (January-April, November-December) and some less (60-70%) during the spring and summer months. Also the amount of FMI-A false LFI detections was increased towards the summer (up to over 40% of the AARI LFI cover in July). For FMI-B the amount of false detection

remained low for the whole year, but the relative amount of FMI-B detections matching with AARI ice chart LFI also decreased towards summer, and was very low in July. On the other hand the total amount of LFI in July was also low and this does not have a large effect on the total classification percentage with respect to the AARI ice chart LFI. The monthly classification results and the relative amount of LFI (monthly fractions of total number of AARI ice chart LFI pixels during the one-year period) are shown in Fig. 8.

For comparing the LFI extent evolution in time the weekly LFI extent over the whole study area during the one-year period corresponding to the AARI ice charts used in this study were computed. These results can be seen in Fig. 9. The FMI-A LFI extent follows the AARI ice chart LFI extent quite well having some larger temporal variations. In the spring the FMI-A LFI extent first decreases some faster (in April 2016) than the AARI LFI but later in the melting period (in May 2016) FMI-A decrease becomes slower compared to the AARI LFI extent. FMI-B systematically gives significantly smaller LFI extent

estimates, approximately 70% of the corresponding AARI and FMI-A LFI extent throughout the whole one year period.

In Fig. 10 LFI extent time series for both FMI-A and FMI-B over the whole study period are shown. According to FMI-A the maximum LFI extent over the study area was around 170000 km$^2$ during the winter 2015-2016 and over 180000 $km^2$ during the winter 2016-2017. The LFI maximum in 2016-2017 was reached later than in 2015-2016. This can partly be explained based on the weather conditions making 2016-2017 a more severe ice winter and having a colder spring than 2015-2016. For

the FMI-B LFI extent time series the LFI extent estimates are approximately 70% of the FMI-A LFI extent and the evolution of the time series is in general similar to that of FMI-A.

The LFI time series were also compared to air temperature measurements at Longyearbyen (see Fig. 1 for the location) weather station in Svalbard (78.22$^o$N, 15.63$^o$E) provided by Met.Norway on http://www.yr,no and to the NCEP/NCAR numerical weather model reanalysis [Kalnay et al. (1996)] air temperature at two locations in the Kara sea area, one near the Kara

gate in the southern Kara Sea and another near the northern tip of the Novaya Zemlya island. The model reanalysis data were

used because there were no weather station measurement data available over the Russian Arctic. The weather data indicated that the winter 2016-2017 was more severe than the winter 2015-2016 at all the three locations. At Longyearbyen the winter temperatures were some milder than for the Kara Sea locations. As seen in Fig. 10 the LFI extent for the winter 2016-2017 was larger than for the winter 2015-2016. Even though the LFI extent can not be explained by the air temperature alone [Olasen (2016)] this comparison between the two winters is in agreement with the air temperature data.

The LFI extent of the sub-regions of southwestern Kara Sea, northeastern Kara Sea and Gulf of Ob are also shown in Fig. 10. The fast ice grows most rapidly in the Gulf of Ob and quite slowly in the southwestern Kara Sea during both 2015-2016 and 2016-2017. Also the LFI melt in the Gulf of Ob is fast when compared to the other two areas presented. This is probably due to the flowing water coming along the river Ob. These annual variations can be compared to those of Fig. 2 in [Divine et al. (2004)].

Also the LFI extent temporal fraction (in percents) for FMI-A and FMI-B at each grid cell over the one year period from November 2015 to October 2016 was computed in a similar manner as in [Fraser et al. (2012)]. For comparison the corresponding fraction was computed also for the weekly AARI ice chart LFI extent of the same one year time period. These numbers also indicate the annual duration of the LFI at each grid cell and are given as percentages of the one year time period in Fig. 11. The results for the whole time period from October 2015 to August 2018 (not shown here) were quite similar, except that the percentages were some higher because part of the summer period of 2017 (with no or very little LFI) was not present in the time series covering the whole study period. The results of the AARI ice chart LFI extent and FMI-A LFI extent were quite similar. In some areas there were minor differences but in general AARI ice chart LFI fraction and FMI-A LFI fraction were in good agreement. On the other hand, FMI-B with the parametrization used here (optimal thresholds $T_{HH}$ and $T_{HV}$) underestimated the LFI extent fraction compared to AARI ice chart LFI extent, but still the same LFI areas were captured, only with a shorter duration of LFI according to FMI-B.

## 5    Discussion and Conclusions

In this study an algorithm for detecting LFI over a test area in the Kara and Barents seas using daily SENTINEL-1 dual-polarized SAR mosaics was developed and tested. Both SAR channels (HH and HV) were used jointly in this study. Two versions of the algorithm were presented: FMI-A applies the optimal thresholds for SAR HH and HV channels and after some post-processing combines the channel-wise LFI estimates, in FMI-B an additional multi-temporal logical operation (logical AND) is performed to decrease the misclassifications to the LFI class at the expense of less detected LFI compared to the AARI ice chart LFI. FMI-B can be considered as an algorithm locating only the areas which very likely represent LFI. Daily LFI extent estimates for a period from October 2015 to August 2017 were generated. The results were also evaluated against weekly Russian AARI ice charts and the correspondence was found to be at an acceptable level, especially when comparing the LFI extent time series. The ice extent over the study area given by the proposed algorithm FMI-A agrees quite well with the AARI ice chart ice extent. However, there exist some differences in the location of the LFI, especially near the detected LFI zone boundaries when compared to the AARI ice chart LFI. During the melting season the difference between AARI ice chart

LFI and FMI-A LFI is some larger than during the freeze-up and mid-winter periods. This can be seen in both the classification error and the ice extent time series. It should be noted here that AARI ice charts are typically based on satellite image analysis (SAR and optical/IR) of a few days prior to the assigning date of the ice chart and the proposed algorithms use SAR data over a two-week period prior to the issuing date. Due to the different time spans and spatial resolutions of the input data sets

differences between the proposed LFI estimates and AARI ice chart LFI may appear.

In this study the optimal threshold based on a training set consisting of AARI ice chart LFI data were used for both FMI-A and FMI-B. During the winter months (November-April) the amount of correctly detected LFI compared to the AARI ice chart LFI was around 80% and the amount of false detections was 20% or less. In the melt period (May-July) the detection rates were worse. By using these settings FMI-B had a good performance in the sense that it makes very little false LFI detections

compared to AARI ice chart LFI. However, then only a little over half of the total AARI LFI is detected by FMI-B. In this sense FMI-A performs significantly better. However, with lower values for the thresholds $T_{HH}$ and $T_{HV}$ FMI-B detects more AARI LFI at the expense of more false LFI detections. For example with the threshold values $T_{HH} = 0.19$ and $T_{HV} = 0.15$ the 68.6% of the AARI LFI is detected and 37.5% of additional (to AARI) LFI were detected by FMI-B.

According to this study the proposed algorithm (FMI-A) is considered suitable for operational LFI detection to be included

in the daily automated FMI sea ice products [Simila et al. (2016); Makynen and Karvonen (2018)] which have this far already been run in an operational test mode over the studied area during a few winters. There also exist plans at FMI to replace the TOPAZ-4 ice model [Sakov et al. (2012)] data used as background information for ice thickness estimation from EO data by the coarse-scale ice thickness from radar altimeter data and by the FMI thermodynamic ice model HIGHTSI over the static ice areas during the static ice time periods detected by the proposed LFI algorithm. For these reasons it is important to get the LFI

information in a similar temporal and spatial resolution as the other sea ice products are. The automated LFI detection also produces systematic LFI analyses which are not dependent on possible subjective interpretation or varying skills of different ice analysts making the ice charts.

Use of temporal cross-correlation average and temporal cross-correlation median in the algorithm produced quite similar results, and because of its faster computation temporal cross-correlation average was selected here instead of temporal cross-

correlation median (requiring sorting of the samples). The execution times on a single CPU-core (Intel Xeon 2.5GHz) with a sufficient amount of RAM memory were reasonable also for operational purposes: computation of LFI extent for one daily SAR mosaic takes 2-4 minutes. As the computation can easily be parallelized, LFI extent estimation e.g. for the whole Arctic or Antarctic areas can easily be performed in a reasonable time by dividing the workload to multiple CPU cores. Here a simple mask based on iteratively estimated distance from land was used to reduce the area of computation and thus fasten the algorithm

execution. This proved to be a very useful feature in computing of longer time series; the execution time was reduced to less than half of the execution time without using the mask.

Compared to other LFI detection methods the proposed method has some advantages. The obvious advantage of using SAR data instead of radiometer data [Seluyzhenok (2011)] is the significantly higher resolution, and the advantage of using SAR data instead of optical or infrared/near-infrared satellite data, such as MODIS [Fraser et al. (2011)], is the ability of SAR to

measure in all weather and lighting conditions independently of clouds or sunlight. LFI can be derived from ice drift based on

multi-temporal SAR imagery [Karvonen (2012)] but estimation of the ice drift is a much more time-consuming process than applying direct temporal cross-correlation. The LFI estimation accuracy of methods based on SAR ice drift is approximately similar to that of LFI-A. Using temporal cross-correlation minimum instead of its average is less robust to local errors than a statistical measure, such as average or median, as it is based only on one value. Methods based on SAR segmentation and SAR backscattering [Eicken et al. (2005); Solomon et al. (2005)] have several potential error sources: C-band SAR backscattering is dependent on the ice surface roughness, which may vary for LFI, SAR backscattering is also dependent on the SAR incidence angle which varies for different acquisitions of a fixed location, and wet snow cover has a significant effect on SAR backscattering making algorithms directly based on SAR backscattering unreliable, especially during the melting season.

An LFI product covering the whole Arctic and Antarctic based on SENTINEL-1 is technically feasible. There is one limitation related to the current SENTINEL-1 acquisition observation scenario: over most of the Arctic SENTINEL-1 is acquiring EW mode HH polarization only and dual-polarized data are acquired only over the European Arctic and Greenland areas. However, it seems that even the HH channel SENTINEL-1 data alone are sufficient for estimating the LFI extent, even though combining the two polarization channels would increase the reliability of the product to some extent. Some preliminary experiments indicated slight increase of false LFI detections compared to AARI ice charts when using HH channel data alone. On the other hand, the amount of correctly detected LFI remained approximately the same as for the combined HH and HV channel classification. The major obstacle of establishing an operational Pan-Arctic/Antarctic LFI service based on SENTINEL-1 SAR imagery is the vast amount of daily SENTINEL-1 data and the current limited data transmission and storage resources.

According to FMI-A there seemed to be LFI even during the summer (late August). The same areas were classified to LFI by FMI-A in both the summers included in this study. This summer LFI naturally represented a classification error over the study area. Those few areas indicated as LFI by FMI-A were typically areas very close to the coastline, and thus very likely due to inaccuracies in positioning of the land mask, causing land or mixed land/sea pixels to be included in the temporal cross-correlation computation. To exclude these areas the LFI detected by FMI-A in late-August 2016 were filtered out (subtracted) from all the FMI-A and FMI-B LFI extent maps.

Future plans include to continue computing a time series of daily LFI extent over the study area and update the LFI data set twice annually, once in spring and once in autumn. The data will be available for interested parties on request on the FMI FTP site in GeoTIFF (thematic map) and NetCDF (numeric data) formats.

Also ways to further improve the current algorithms need to be studied. Some interesting alternatives for future algorithm development are the use of varying thresholds according to the time of the year or weather data, and applying a dual-threshold temporal cross-correlation average thresholding scheme, i.e. first applying a lower threshold, then applying a higher threshold and finally combining the two results in an optimal way. Also a combination of applying FMI-B and FMI-A could be studied. This combination could first locate the areas representing LFI with a high likelihood (FMI-B) and then extend the this initial LFI extent based on the FMI-A result adjacent to the initial LFI areas detected by FMI-B (a region growing method).

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

**Table 1.** Comparison of the FMI methods to AARI ice charts, the numbers are in percents. The values in parentheses are standard deviations in percentage points of the AARI ice chart LFI.

| Method | Detected (%) | False detection (%) |
|--------|--------------|---------------------|
| FMI-A  | 73.1 (8.8)   | 20.9 (11.8)         |
| FMI-B  | 50.4 (13.2)  | 4.3 (2.2)           |

**Figure Captions**

Figure 1. The study area in the used polar stereographic projection.

Figure 2. SAR mosaics of March 8, 2016, HH mosaic (a) and HV mosaic (b). The land areas appear as green and areas of no data as black in the figures.

Figure 3. Block diagram of the LFI detection (FMI-A) for SENTINEL-1 HH polarization channel. The process for the HV channel is similar, except a threshold value of $T_{HV}$ is applied instead of $T_{HH}$.

Figure 4. Mask used to locate the areas where LFI is searched. White areas indicate the LFI search area, green areas are land.

Figure 5. The total number of erroneously classified pixels as a function of the temporal cross-correlation average for HH channel SAR data (a) and for HV channel SAR data (b). The optimal thresholds were defined as the minimum of the total error ("sum" curves according to the figures legend).

Figure 6. AARI ice chart of March 8, 2016, translated to the polar stereographic projection used in this study and cropped to the study area.

Figure 7. LFI extent based on AARI ice chart (a), FMI-A LFI (b) and FMI-B LFI (c) of March 8, 2016. LFI areas are the black areas in the figures.

Figure 8. Monthly detection and false detection percentages for FMI-A (a) and FMI-B (b) compared to AARI ice chart LFI, and the relative amount of (AARI) LFI points (c) in percents of the LFI points of the whole year.

Figure 9. Ice extent time series of AARI ice charts, FMI-A and FMI-B during the one-year period from November 1 2015 until October 31, 2016. The time series is weekly with FMI-A and FMI-B for the same days as the weekly AARI ice charts.

Figure 10. FMI-A (a) and FMI-B (b) LFI time series for the whole study period from October 15 2015 until August 31 2017. Also the time series of Kara Sea sub-regions, southwestern (SW), northeastern (NE) and Gulf of Ob (Ob) have been included in the figures. The division of Kara Sea into sub-regions (c).

Figure 11. Temporal LFI coverage (percentage) during the period from November 1 2015 until October 31 2016 based on the weekly AARI ice charts (a), daily FMI-A (b) and daily FMI-B (c). The values (percentages) indicate how long time fraction of the one year period there have been LFI at each grid cell.

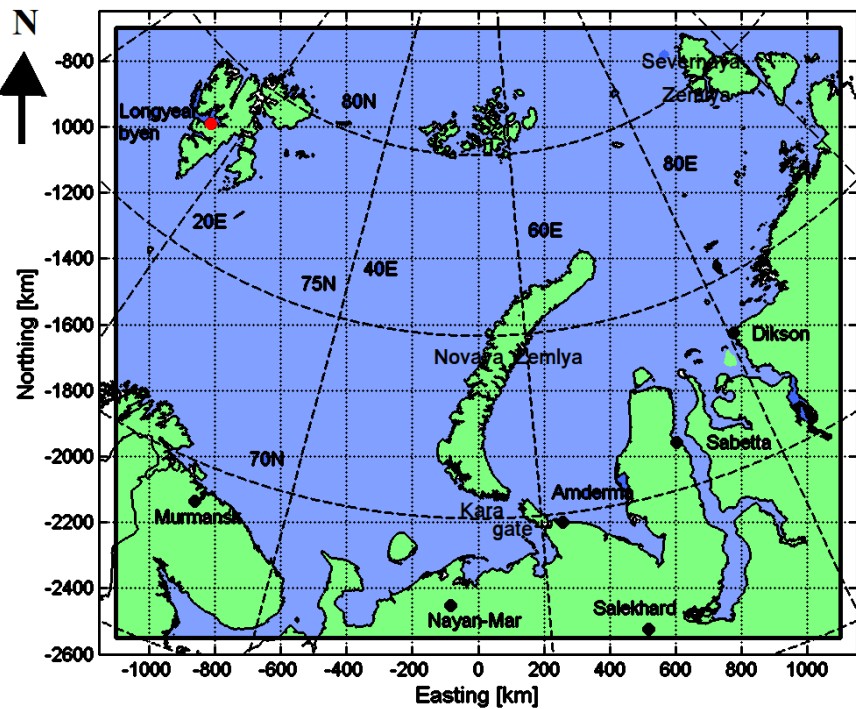

**Figure 1.** The study area in the used polar stereographic projection.

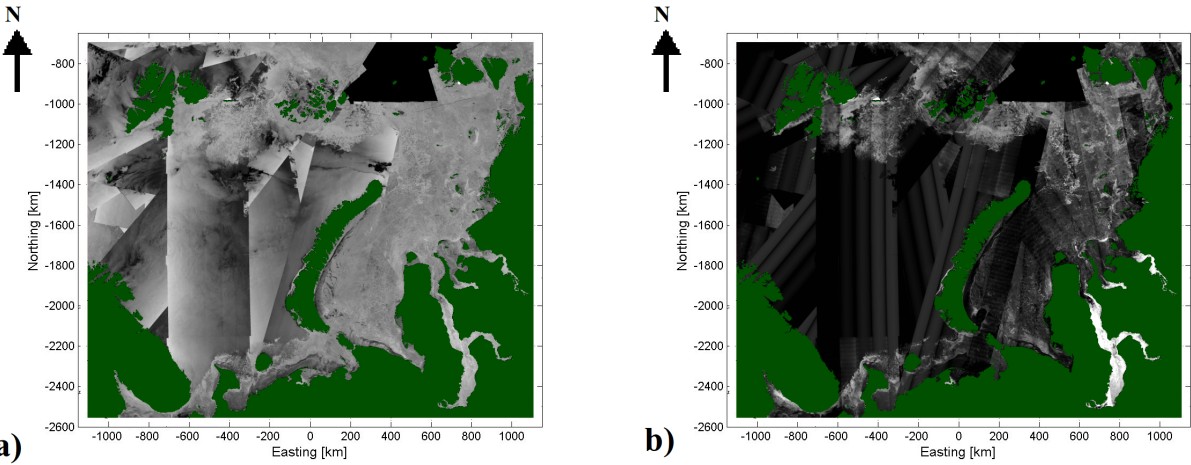

**Figure 2.** SAR mosaics of March 8, 2016, HH mosaic (a) and HV mosaic (b). The land areas appear as green and areas of no data as black in the figures.

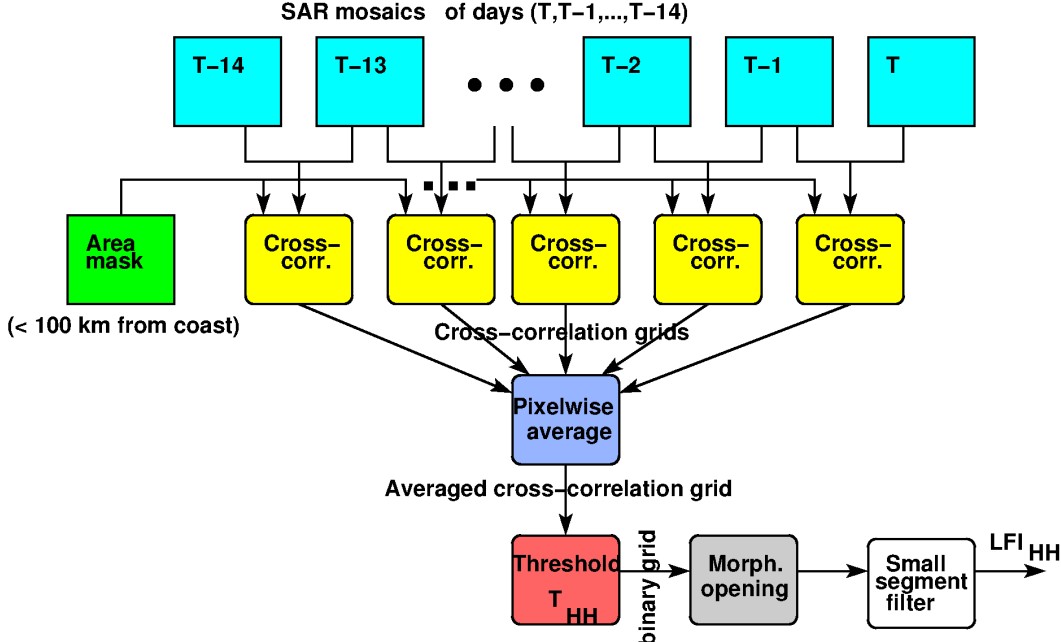

**Figure 3.** Block diagram of the LFI detection (FMI-A) for SENTINEL-1 HH polarization channel. The process for the HV channel is similar, except a threshold value of $T_{HV}$ is applied instead of $T_{HH}$.

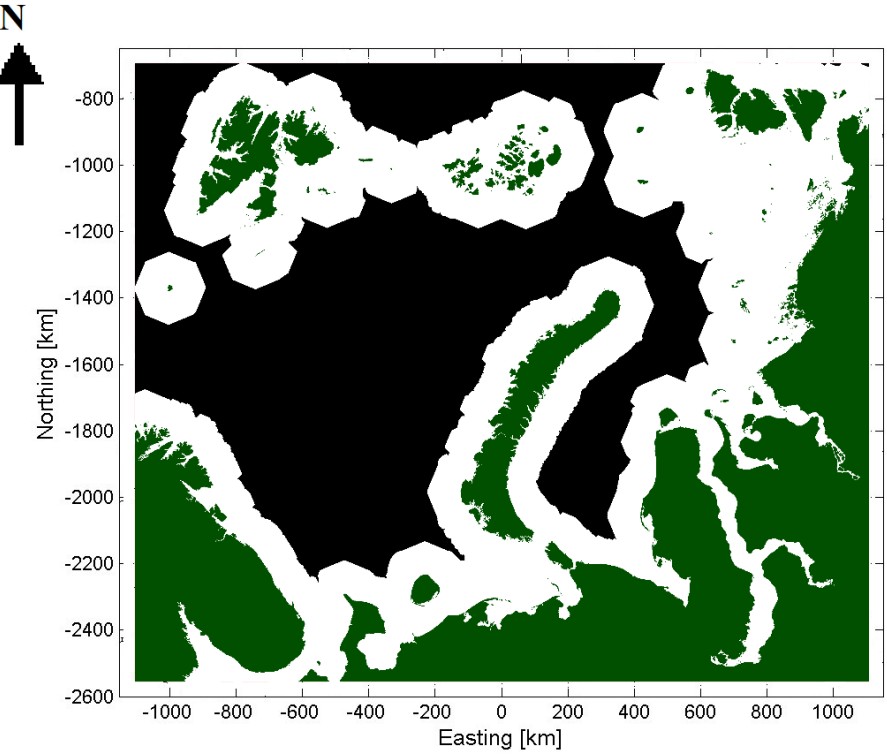

**Figure 4.** Mask used to locate the areas where LFI is searched. White areas indicate the LFI search area, green areas are land.

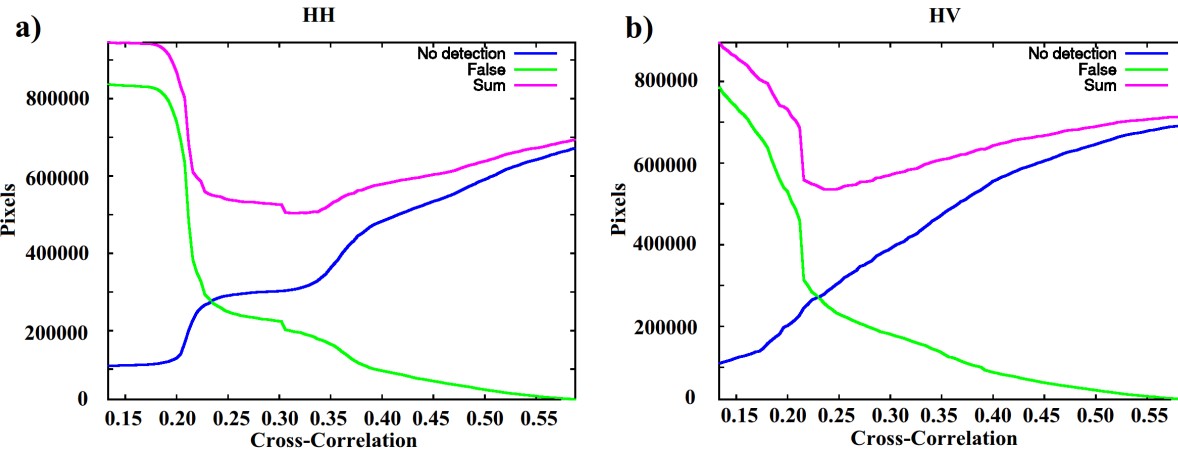

**Figure 5.** The total number of erroneously classified pixels as a function of the temporal cross-correlation average for HH channel SAR data (a) and for HV channel SAR data (b). The optimal thresholds were defined as the minimum of the total error ("sum" curves according to the figures legend).

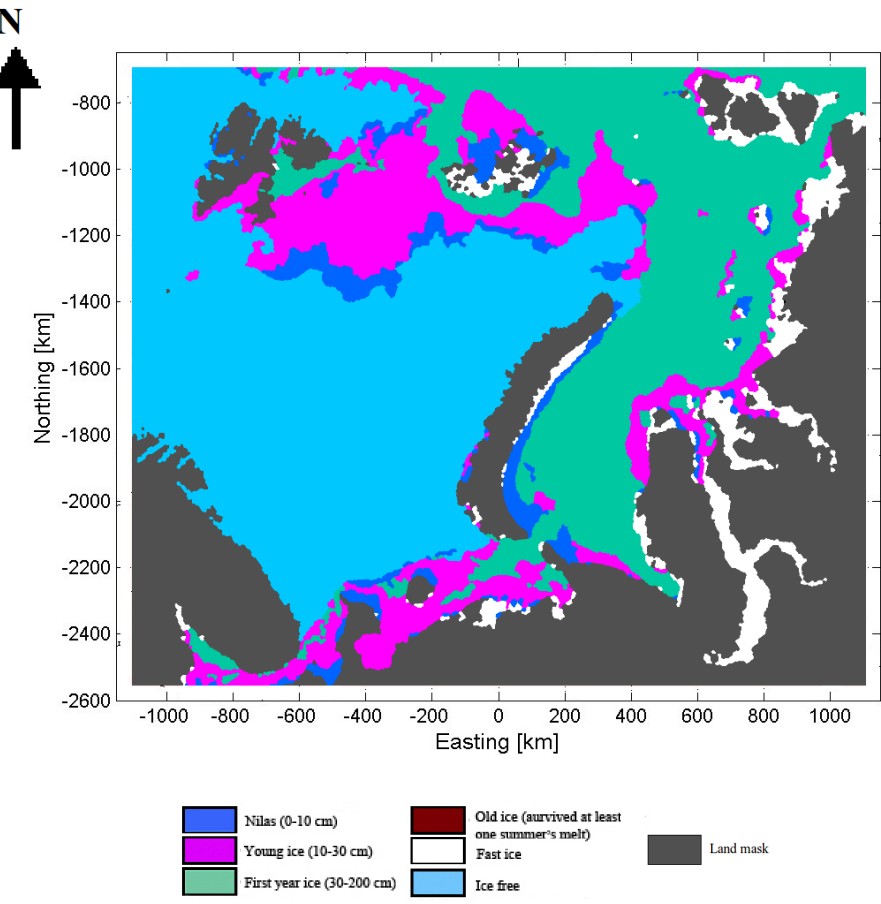

**Figure 6.** AARI ice chart of March 8, 2016, translated to the polar stereographic projection used in this study and cropped to the study area.

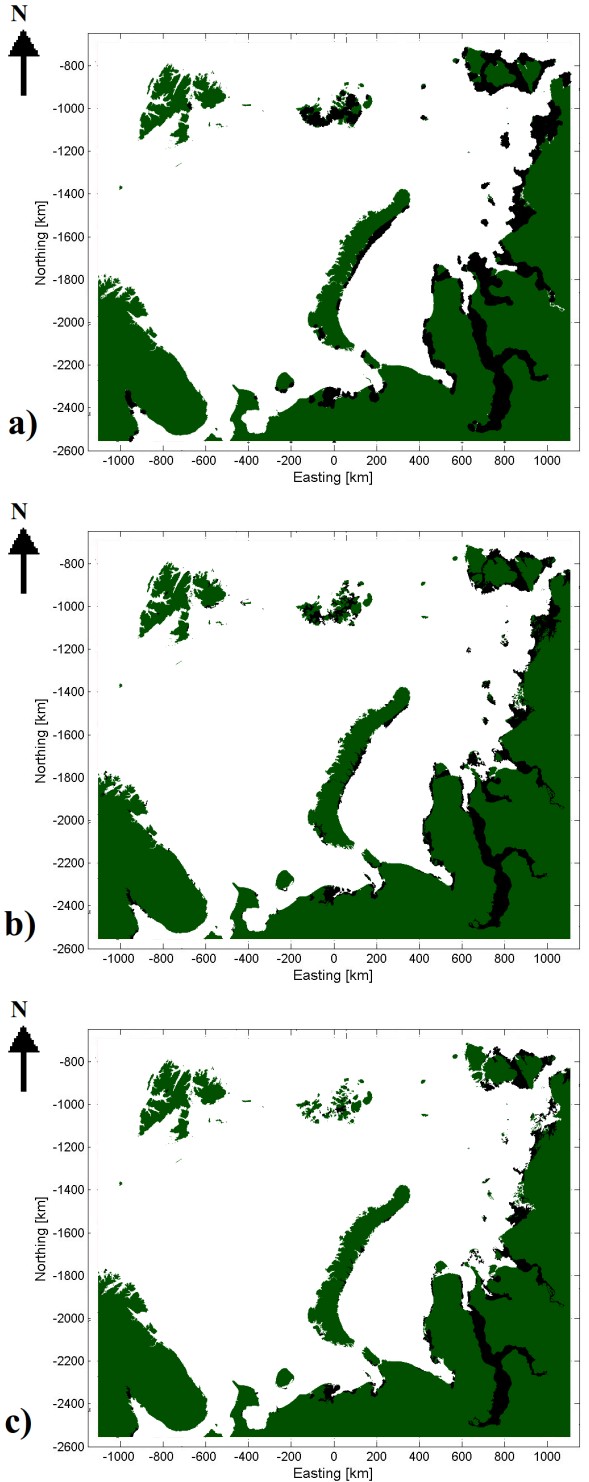

**Figure 7.** LFI extent based on AARI ice chart (a), FMI-A LFI (b) and FMI-B LFI (c) of March 8, 2016. LFI areas are the black areas in the figures.

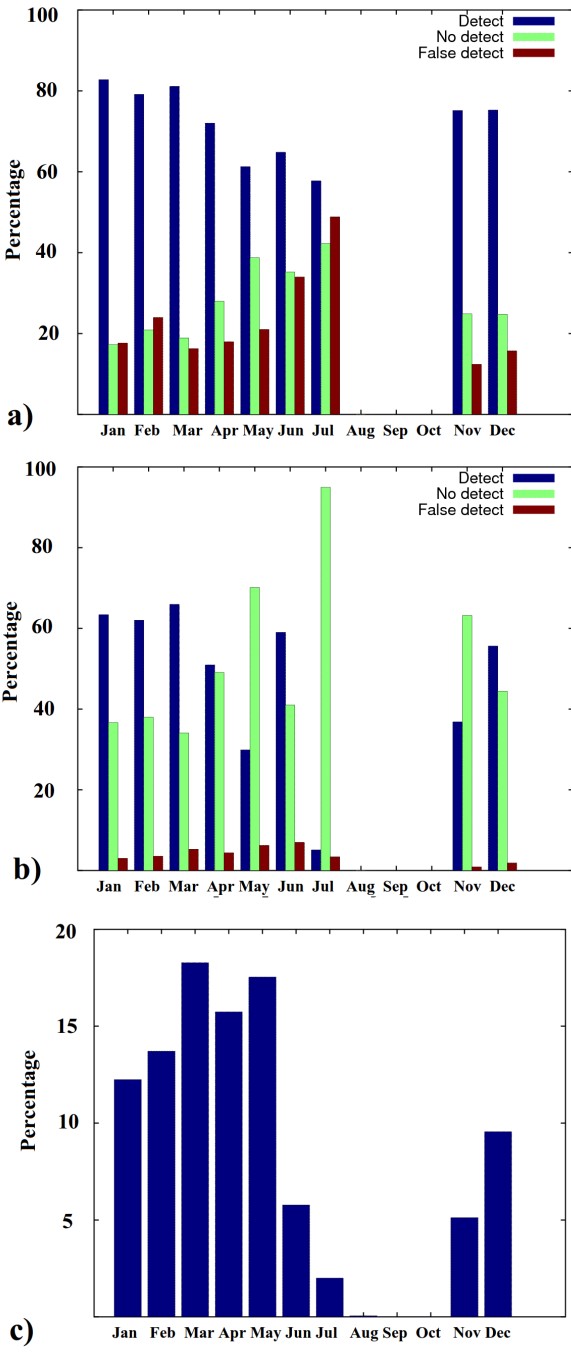

**Figure 8.** Monthly detection and false detection percentages for FMI-A (a) and FMI-B (b) compared to AARI ice chart LFI, and the relative amount of (AARI) LFI points (c) in percents of the AARI LFI points of the whole year. This indicates the relative amount of LFI cover for each month compared to the LFI cover of the whole year.

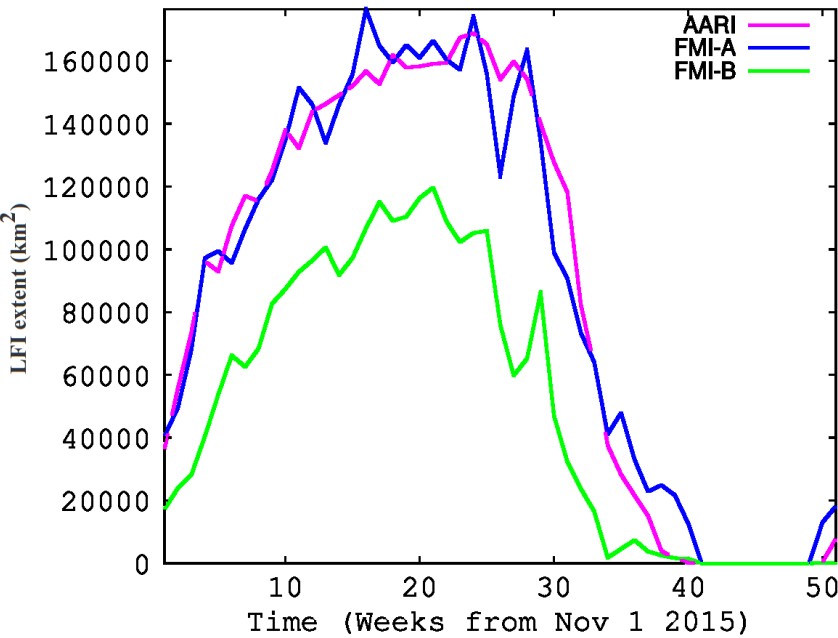

**Figure 9.** Ice extent time series of AARI ice charts, FMI-A and FMI-B during the one-year period from November 1 2015 until October 31, 2016. The time series is weekly with FMI-A and FMI-B for the same days as the weekly AARI ice charts.

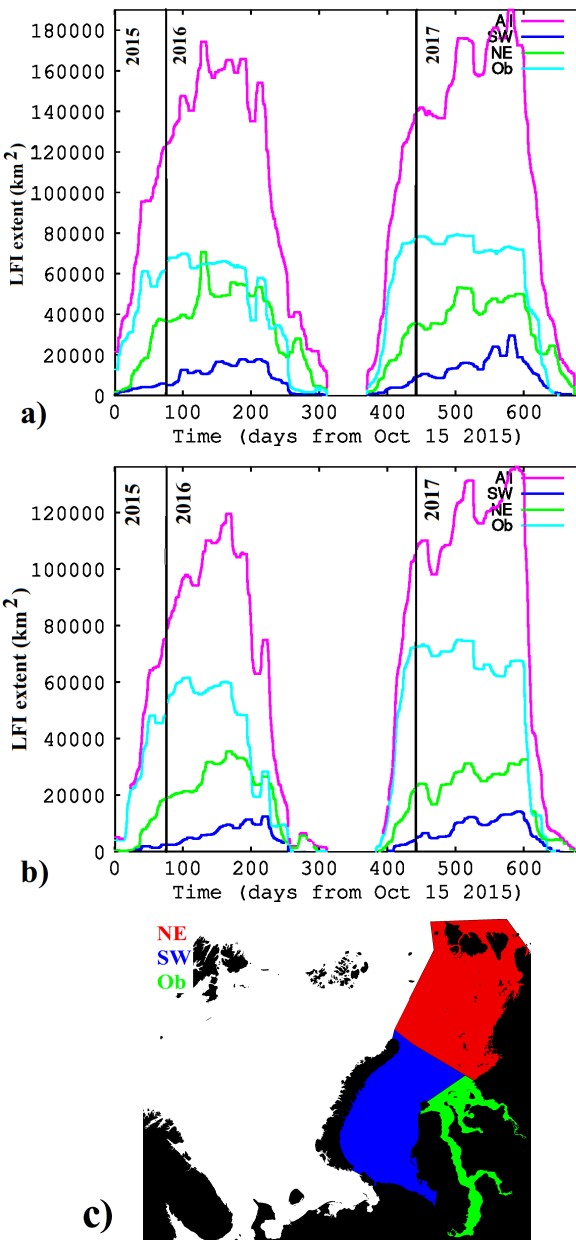

**Figure 10.** FMI-A (a) and FMI-B (b) LFI time series for the whole study period from October 15 2015 until August 31 2017. Also the time series of Kara Sea sub-regions, southwestern (SW), northeastern (NE) and Gulf of Ob (Ob) have been included in the figures. The division of Kara Sea into sub-regions (c).

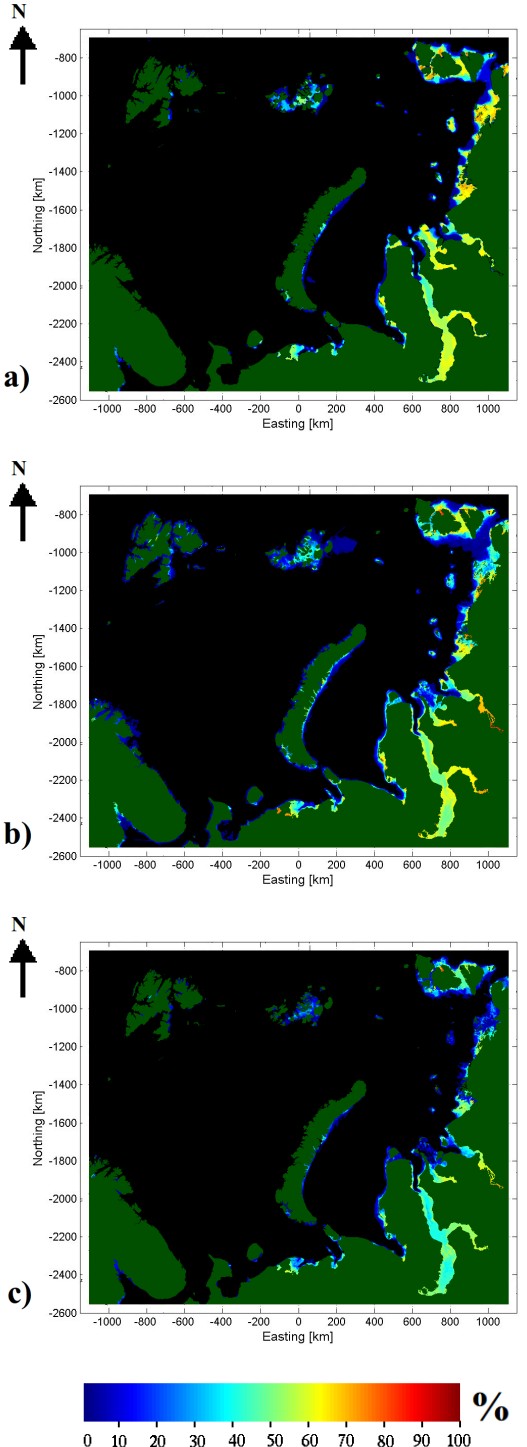

**Figure 11.** Temporal LFI coverage (percentage) during the period from November 1 2015 until October 31 2016 based on the weekly AARI ice charts (a), daily FMI-A (b) and daily FMI-B (c). The values (percentages) indicate how long time fraction of the one year period there have been LFI at each grid cell.