# Peer review of "Estimation of Arctic Land-Fast Ice Cover based on Dual-Polarized SENTINEL-1 SAR Imagery"

_The Cryosphere, 2017_

## Referee Comment (RC1) · Anonymous Referee #1 · 4 Jan 2018

General comments: This manuscript is a nicely self-contained "techniques" style paper describing a new method for generating maps of landfast sea ice from SENTINEL-1 EW imagery. It provides an effective overview of the significance of fast ice and the field of/recent progress in fast ice detection from satellite sensors. The author's use of English is not perfect, but the meaning was perfectly clear in almost all cases. Unfortunately the large number of formatting errors (erroneous spaces and parens in references; missing references; missing figure numbers; typographical error in the abstract) detracted a little from the presentation, but these can be fixed easily.

Specific comments: 1) This paper is possibly of too limited scope to be considered to be considered for publication in The Cryosphere. As it stands, it is a nice "techniques" style paper, but there is little in the way of scientific results (a very short time-series

of fast ice extent; and a snapshot of fast ice retrievals from AARI charts vs the two algorithms presented here, neither of which are explored in any detail). I suggest that the Editor consider whether such a techniques-focused paper is suitable for publication here. Alternatively, the author may significantly increase the amount of analysis of the data here, or expand its scope (spatially). 2) The paper claims that "we have used quite similar criteria" to Mahoney et al., 2005 for LFI. Notwithstanding the use of "we" for a solo author, this claim is hard to support. You say that you use similar criteria to Mahoney et al's "contiguity" and "20 day" criteria. Neither appears to be true: a. Nothing in your methods description suggests that you enforce contiguity in any way; and b. Your FMI-A and FMI-B algorithms use a two day and 14 day time-scale for fast ice classification, respectively. FMI-B could easily have been changed to a 20-day criterion if you really wanted to be similar to the Mahoney work. I'm not suggesting that your failure to have similar criteria to Mahoney et al is a bad thing – in fact, at least in the Antarctic, contiguity with the coast is certainly not necessary for fast ice formation, and the 20 day figure suggested by Mahoney is certainly up for discussion. I only take issue in the fact that you suggest you are enforcing similar criteria whereas you certainly aren't. 3) Section 3, the methodology, is lacking in clarity. a. P3,L26: I don't believe the mask is 100 km from the nearest coast, as you say. E.g., see Fig 5 – a polygonal shale is clearly seen around small islands, probably the consequence of mathematical morphology operators for a certain number of pixels – and not a 100 km distance from nearest coast, as stated. b. As you state, an ice mask based on bathymetry would be so much better, and almost as easy to implement. Are there any regions of fast ice that would have been identified more than 100 km offshore, which you are ignoring here because of this? c. P3,L30 to P4,L1: Aren't you using daily mosaics, meaning that a cross-correlation of r=1 between daily mosaics would be impossible? I suspect your description of mosaic construction is lacking. How many days can scenes get "reused" for? Doesn't this temporally smear your result, especially in the case of FMI-A? d. There is no mention of how the TCC thresholds were decided upon. What's the sensitivity of the choice? e. Similarly, there's no

description of the choice of 14 days for the FMI-B algorithm. 4) P1,L17: You need to explicitly mention which region these statements are true for. 5) P2,L21: Conspicuous absence of the Meyer et al reference here - doi:10.1016/j.rse.2011.06.006 6) P2,L28-31: Unnecessary detail for a manuscript – consider removing 7) P5,L13: "rather good" is not quantitative enough – stemming from the rather qualitative comparison between datasets. 8) P6,:15: This temporal average and temporal median is introduced here (discussion and conclusion) for the first time, but I have no idea what it's referring to. This whole section (on processing time) seems excessively lengthy. 9) P5,L21-25: You state that HH alone would be sufficient for the techniques here, but there is no evidence to back this statement up. 10) P5,L26-31: A repeat of earlier in the paper. 11) You say how quick this algorithm runs. Why do you need to use a LFI search area mask then? Why not run it everywhere in the ocean? 12) I'm left with a lingering desire for a map of average fast ice coverage, even though your time series is very short. This would be a nice result for this paper, and allow comparison to the earlier Divine work. Somethin akin to the Fraser et al., 2012 figure 2 (https://doi.org/10.1175/JCLI-D-10-05032.1) would be ideal.

Table 1: I have no idea what the column headings represent. (A(IC), etc.) This whole table is very poorly described. Figure 1: This map is poorly presented, and appears to be of low resolution. Borderline illegible. Figure 2: Completely illegible. Figure 3: The choice of a black mask is not appropriate given the high amount of black in the right sub-figure. The figure needs to be made bigger. An overlay needs to be included (lat/lon, coastlines, etc.) Figure 4: Caption doesn't indicate that this is for FMI-B. I question the necessity of this figure too. Figure 6: The caption should refer to FMI-A and FMI-B. Again, overlay needed (lat/lon/colour legend for upper left). This figure seems redundant with Fig 7 also included. Figure 7: Need a legend describing all 8 potential colours used in this figure. Also overlay, etc. Figure 8: No comment on why the AARI charts underestimate fast ice compared to your work. Figure 8: Doesn't the value of ∼0 km^2 fast ice at day 180 in the FMI-B algorithm indicate that your coast mask is just fine? This is contrary to your statement in the text that some land pixels

are not correctly masked out.

Originality (novelty): - Yes – new methods, insights and data.

Scientific quality (rigour): - Purpose of the work is clearly articulated, but the description of the methodology is borderline poor. Techniques appear to be valid and suitable. Results are limited – this appears to be mainly a techniques paper with basic validation. Consideration of other work is up to scratch though.

Significance (impact): - In current form, significance is limited. It's a very short time series covering a relatively limited geographic region (despite the grandiose title!). It is a good "techniques" paper but there is scant exploration of the time series presented. Given its limited scope, I question its suitability for TC.

Presentation quality: - The presentation quality is not very good. Typographical errors throughout, references missing, references not presented well, etc. Figures are lacking clarity, size, and essential elements. English use is quite strange in parts but I unfortunately don't have time to go through it in more detail.

In the full review and interactive discussion, the referees and other interested members of the scientific community are asked to take into account all of the following aspects: 1. Does the paper address relevant scientific questions within the scope of TC? Y 2. Does the paper present novel concepts, ideas, tools, or data? Y 3. Are substantial conclusions reached? N 4. Are the scientific methods and assumptions valid and clearly outlined? Y 5. Are the results sufficient to support the interpretations and conclusions? Y 6. Is the description of experiments and calculations sufficiently complete and precise to allow their reproduction by fellow scientists (traceability of results)? N 7. Do the authors give proper credit to related work and clearly indicate their own new/original contribution? Y 8. Does the title clearly reflect the contents of the paper? N 9. Does the abstract provide a concise and complete summary? N 10. Is the overall presentation well structured and clear? N 11. Is the language fluent and precise? N 12. Are mathematical formulae, symbols, abbreviations, and units correctly defined and used?

N/A 13. Should any parts of the paper (text, formulae, figures, tables) be clarified, reduced, combined, or eliminated? Y – method section needs clarification 14. Are the number and quality of references appropriate? Y 15. Is the amount and quality of supplementary material appropriate? N/A

―――――――――――――――

---

## Referee Comment (RC2) · V. Selyuzhenok (Referee) · 20 Jan 2018

General Comments:

The manuscript presents a new method for the landfast ice mapping based on SENTINEL-1 SAR imagery. The method is tested in the Kara and Barents Sea area and the resultant landfast ice (LFI) product compared to operational sea ice charts from Arctic and Antarctic research Institute, Russia (AARI charts). The results indicate the potential to derive reliable fast ice extent operationally. Unfortunately missing methodological details, inconsistent results and the large number of typographical and formating errors do not make an impression of a self-contained manuscript.

Specific comments:

**1. Missing discussion of other relevant studies**

The introduction give an overview on of existing methods of fast ice detection, including several SAR-based methods. However, it is not clear what is the potential advantage of the proposed algorithm compared to the existing ones.

p1, line 21-22: The author states: "in the fast ice zone only thermodynamic ice modeling is necessary as the modeled dynamic part can be omitted". Fast ice can be formed dynamically, it also can breakup in response to the dynamical forcing. Please, clarify your statement. It would be good to provide some references to model studies to support your statements on p 1, line 21-23.

**2. Missing clarity in the methodological sections**

In general, the description of the work-flow is confusing. It needs to be clarified in order to be reproducible:

Were raster (gif, png) or vector (shp) AARI charts used? In general, the vector format is more convenient to work with. Fig 2. and Fig 6. (upper left) suggest that the raster format was used. What kind of software was used to re-project the rasters and extract fast ice extent?

I encountered that different product might give quite variable fast ice extent during its development in fall. It would be interesting to see whether the FMI methods show result in similar to the AARI fast ice extent in October-November. The Sentinel-1-based time series of LFI extent were derived from October 2015, however the comparison with the AARI charts covers a period from November on. What is the reason for shortening the comparison period?

p 4, line 16-17: The author should clarify what is "the daily LFI product". Is it a product of FMI-A method? "Daily mosaics" seems to contain SAR data collected for a period of several days. The consecutive mosaic might have several days apart. In this case, the LFI product contains information from different dates.

The construction of "daily mosaics" is described in 2.3 (p 3, lines 12-14). Later it becomes clear (p 3, lines 30-31) that some parts of a daily mosaic might remain from a previous day. The author should explain better how and for what time period the mosaics are constructed.

The use of 0.19 and 0.15 TCC threshold values are not explained. In general, the values seem to be rather low. According to Fig. 6, the threshold values work well for discriminating between fast ice and open water or newly-formed ice, but appear to work worse when fast ice is surrounded by the first-year ice (area south-east of the Severnaya Zemlya Archipelago). Were the thresholds picked based on a sensitivity study?

p 4, line 16-17: The author should clarify what is "the daily LFI product". Is it a product of FMI-A method? "Daily mosaics" seems to contain SAR data collected for a period of several days. The consecutive mosaic might have several days apart. In this case, the LFI product contains information from different dates.

p 5, line 15. The temporal average and temporal median are mentioned for the first time. Please, provide more information in the methodology and results sections.

p 5, lines 26-31: Application of an extended land mask would help to properly compare your results with AARI charts. First, it would exclude summer fast ice. Second, it will reduce the difference between AARI and FMI fast ice extent (in Fig. 8) and therefore add more value to the number describing the remaining differences. I suggest that the extended land mask should be applied at least for the data comparison.

3. Inconsistency of the results

p 4, lines 23-24: The description method performance does not agree with figures and table. The qualitative description "quite good" is not in line with the quantitative characteristics presented in Table 1. According to the Fig. 8, FMI methods slightly overestimate fast ice area compared to the AARI charts. If I understand the numbers

in Table 1 correctly, at least half (92.+-42.3% ) of fast ice detected by FMI-A method is not actually fast ice on the AARI charts. A large area of spurious fast ice is located far from the coast between the Severnaya Zemlya and the mainland (Fig. 6, 7). Its presence can not be explained by different land mask, as suggested on p 4, line 27-28. The author claims that his methods are more precise than the AARI charts. Currently, operational charts is the most consistent and reliable source of information on fast ice extent. A cross-comparison of two data sets does not reveal a more precise method, but rather gives information about relative performance of the two. Fig. 6, 7 show that some fast ice areas (FMI-A and FMI-B) are detached from the coast, which suggests that automated methods are less precise, than the AARI charts. As the author says, different fast ice definition may indeed explain mismatch between data sets. The author however should explain what are the differences in fast ice definitions and how they may affect fast ice detection process. The given definition: "our automated algorithms locate the ice areas which has been static over a given time" is misleading. The presented method is based on reveling areas of low changes in surface backscatter characteristic, which does not ultimately mean that the ice (or any other surface) was motionless. It is also not clear what is the "given time".

LFI area from FMI-B methods differs on Fig. 8 and Fig. 9. On Fig. 8 the maximal FMI-B LFI extent is reached between Julian days 100 and 120 (April-May); it is roughly 35 000 km2. The LFI extent for the same time period on Fig. 9 (170 - 200 days from Oct 15, 2015) exceeds 40 000 km2. Which of the figures is correct?

4. Questionable conclusions

p 5, lines 13-14: The author states that the method is suitable for operational LFI monitoring. Taking into account presence of large areas of spurious fast ice (Fig. 6, 7), inconsistent fast ice extent presented in Fig. 8, 9 and results from pixel-wise comparison with AARI charts (Table 1), I question that at this stage the methods can be use for reliable fast ice detection.

5. Figures and table require a better explanation

The technical information shown in Fig. 1, 2, 5, 6, 7 can be presented more efficiently. E. g. the the study area (Fig.1) and the land mask and LFI mask (Fig. 5) can be shown in one figure. The AARI ice chart (Fig. 2) is duplicated in Fig. 6 (upper right corner). Fig. 6, 7 show the same information. Table 1 is poorly explained. Please name the steps in the flowchart (Fig. 4) in consistency with the text. E. g. Cross-corr. Is TCC in the text; area mask is referred as a mask in the text. What doest Pixelw average stand for? Please, describe in the text. All figures require better captures, legend, geographic information and land mask (if applicable).

Technical corrections:

p 1, line 2: Please replace "ove Kara and Barents Seas" by "over the Kara and Barents Seas"

p 1, line 8: Please remove excessive spaces before commas in citations here and throughout the text

p 1, line 11: Missing citation after Zubov, 1945

p 1, line 12-13: Do Yu et al. (2014) indeed give this number in their paper? Please, rephrase, in case the 13% is not mentioned by Yu et al. (2014).

p 1, line 13: Please replace "sea ice coverage" by "sea ice cover"

p1, line 16: "quite similar criteria" is kind of vague. Please clarify.

p 1, line 20: Wrong citation. To support your statement, use the work by Maqueda, M., Willmott, A.J. and Biggs, N.R.T., 2004. Polynya dynamics: a review of observations and modeling. Reviews of Geophysics, 42(1). The importance of fast ice was not studied by Selyuzhenok et al. (2015). The paper rather describes changes in the fast ice regime. Please, move the reference to p 1, line 18 : "later formation and earlier disappearance (Mahoney et al. , 2014, Selyuzhenok et al., 2015)".

p 1, line 24-25: "The proposed method has been used and will be used for. . ." Has the method been used before? The sentence seems to be out of the context. Please move it to the end of the introduction, where the proposed method is introduced.

p1, line 25: What is the existing LFI time series? Are you referring to the AARI charts or another product? Please clarify.

p 2, line 8: Please replace " in the case on" by " in the case of"

p 2 line 14-15: The sentence starting with "In Mahoney et al. 2004, 2005.." sounds as the fast ice was detected based on mosaic edge, orientation and temporal difference. I suggest changing to "In Mahoney et al. (2004, 2005) LFI is detected based on vector grayscale gradient fields of 3 subsequent SAR images"

p 2, line 2: Replace "re-reprojected" by "reprojected"

p 3, line 21: adjacent daily SAR mosaic?

p 3, line 23: in Fig 4.?

p 3, line 25: To increase computing performance and to exclude. . . ?

P 3, line 29: Please replace "i,e, white areas in Fig. 4" by "i. e. white areas in Fig. 5"

p 4, line 1: " less than zero", Did you mean "less than one" or it is indeed negative?

P 4, line 8: Please remove "still" in " We still additionally applied.."

p 4, line 12: Please remove "still" in " we still additionally perform.."

p 4, line 12: Please replace "logical and operation" by "logical AND operation"

p 4, line 13: Please remove "in this context"

p 4, line 15: Typo in "results"

p 4, line 19-22: Inconsistent terminology: FMI algorithms, SAR algorithms

p 4, line 25: Missing figure number (7)

p 4, line 32-33: Duplicated "whole study" and "our study area"

p 5, line 3: Typo in "erroneous"

p 5, line 10: Typo in "developed"

---

## Author Comment (AC1) · 5 Mar 2018

General comments: This manuscript is a nicely self-contained "techniques" style paper describing a new method for generating maps of landfast sea ice from SENTINEL-1 EW imagery. It provides an effective overview of the significance of fast ice and the field of/recent progress in fast ice detection from satellite sensors. The author's use of English is not perfect, but the meaning was perfectly clear in almost all cases. Unfortunately the large number of formatting errors (erroneous spaces and parens in references; missing references; missing figure numbers; typographical error in the abstract) detracted a little from the presentation, but these can be fixed easily.

Thank You for Your comments. I have tried to improve the language, improve the

formatting and checked the references and figure numbers. This version has done through a lot of changes from the previous version. If seen necessary I can still try to improve the English language and also if seen neccessary find asuitable perosn to make an English language proofreading.

Specific comments:

1) This paper is possibly of too limited scope to be considered to be considered for publication in The Cryosphere. As it stands, it is a nice "techniques" style paper, but there is little in the way of scientific results (a very short time-series of fast ice extent; and a snapshot of fast ice retrievals from AARI charts vs the two algorithms presented here, neither of which are explored in any detail). I suggest that the Editor consider whether such a techniques-focused paper is suitable for publication here. Alternatively, the author may significantly increase the amount of analysis of the data here, or expand its scope (spatially).

The S-1 dual-polarized data covers only the European Arctic and in other areas the S-1 data are in single-polarization (HH) mode. In principle the spatial scope could be increased to cover e.g. Greenland waters. However, we have collected the data and generate mosaics only over the area used in this study. Collecting the data even over all the European Arctic (from ESA SENTINEL SciHUB) would be an enormous job and the amount of data too much for DMI data handling and storage capasity. For this reason the study area has been restricted as it is. For climate studies the time series is still short, and the major purpose of this manuscript has been to demonstrate the capability of the proposed methodology method and S-1 data for LFI monitoring and also to start a LFI time series over the study area. For a SAR related manuscript the amount of data used is exceptionally large.

There have been many papers in TC with a significantly smaller amount of data and significantly shorter time period. One example of such paper is: The Cryosphere, 12, 343-364, 2018 https://doi.org/10.5194/tc-12-343-2018

The major target of this manuscipt indeed is to represent the algorithms and demonstrate their capability. But also some analysis for the study area and the study period have also been included.

In addition it should be noted that the amount of data used for this study is exceptionally large, several thousands of SAR images were used to generate the mosaics even though the time period is less than two years. Actually many SAR-related papers only study a few images.

2) The paper claims that "we have used quite similar criteria" to Mahoney et al., 2005 for LFI. Notwithstanding the use of "we" for a solo author, this claim is hard to support. You say that you use similar criteria to Mahoney et al's "contiguity" and "20 day" criteria. Neither appears to be true: a. Nothing in your methods description suggests that you enforce contiguity in any way; and b. Your FMI-A and FMI-B algorithms use a two day and 14 day time-scale for fast ice classification, respectively. FMI-B could easily have been changed to a 20-day criterion if you really wanted to be similar to the Mahoney work. I'm not suggesting that your failure to have similar criteria to Mahoney et al is a bad thing – in fact, at least in the Antarctic, contiguity with the coast is certainly not necessary for fast ice formation, and the 20 day figure suggested by Mahoney is certainly up for discussion. I only take issue in the fact that you suggest you are enforcing similar criteria whereas you certainly aren't.

I included the adjacency to land condition in the algorithms and the results have been reproduced. I still use the 14 day period instead of 20 days as the results are quite similar (based on our experiments with the Baltic Sea LFI) but with a shorter time period the temporal resolution is better. "quite similar" has been removed.

3) Section 3, the methodology, is lacking in clarity. a. P3,L26: I don't believe the mask is 100 km from the nearest coast, as you say. E.g., see Fig 5 – a polygonal shale is clearly seen around small islands, probably the consequence of mathematical morphology operators for a certain number of pixels – and not a 100 km distance from

nearest coast, as stated. b. As you state, an ice mask based on bathymetry would be so much better, and almost as easy to implement. Are there any regions of fast ice that would have been identified more than 100 km offshore, which you are ignoring here because of this? c. P3,L30 to P4,L1: Aren't you using daily mosaics, meaning that a cross-correlation of r=1 between daily mosaics would be impossible? I suspect your description of mosaic construction is lacking. How many days can scenes get "reused" for? Doesn't this temporally smear your result, especially in the case of FMI-A? d. There is no mention of how the TCC thresholds were decided upon. What's the sensitivity of the choice? e. Similarly, there's no description of the choice of 14 days for the FMI-B algorithm.

The mask has been generated iteratively from the GSHHG coastline, so each pixel adjacen to coastline are first given a distance from the coastline (either 1 ot sqrt(2) depending on whether they are horizontal/vertical or diagonal neighbors. Then this in applied incrementally to the neighboring pixels of the pixels with a distance from the coastline assigned until diustances from coastline for the whole area have been assigned. This description has been included. Even though the distances are not exact, the mask is useful. The aim of the mask is to fasten the computation, actually it does not have any other effect on the results, only the faster computation. The faster execution is important in getting time series computed in a reasonable time (even though for a single day the computation typically only takes a few minutes).

More detailed description of the mosaic construction has been added.

The TCC thresholds in the previous version were based just on experiments and visual interpretation of two (February and April) mosaic images. In the updated version the TCC thresholds were based on comparing the thresholding results to 5 monthly (Jan-May 2016) AARI ice charts and defining the optimal (corresponding to minimal classification error) thresholds by varying the threshold and computing the classification errors. This information with related figures have been included.
4) P1,L17: You need to explicitly mention which region these statements are true for.

I mean Arctic in general, "Arctic" added in the text.

5) P2,L21: Conspicuous absence of the Meyer et al reference here - doi:10.1016/j.rse.2011.06.006

Reference added. Thank you!

6) P2,L28-31: Unnecessary detail for a manuscript – consider removing

I think the exact information of the test area location is important, for example if someone would be interested to compare his/her results to the results in this manuscript. So I did not remove this detail.

7) P5,L13: "rather good" is not quantitative enough – stemming from the rather qualitative comparison between datasets.

The comarison results with the updated parametrization have changed and Table 1 has been simplified and made more clear.

"rather good" replaced by "at an acceptable level".

8) P6,:15: This temporal average and temporal median is introduced here (discussion and conclusion) for the first time, but I have no idea what it's referring to. This whole section (on processing time) seems excessively lengthy.

Changed to temporal cross-correlation average andd median. Temporal average and temporal cross-correlation median refer to averaging or taking median of cross-correlation values at the same grid point in multiple mosaic images representing different days. I hope this has been expressed more clearly in the revised manuscript.

9) P5,L21-25: You state that HH alone would be sufficient for the techniques here, but there is no evidence to back this statement up.

It has not been studied comprehensively yet as in this area we have HH/HV data.

Some tests with Baltic Sea ice have given promising results for HH alone. I have tried to reprase this sentence. Unfortunately no real numerical comparison to support this assumption yet and not much resources for the work. Only some visual comparisosn have been made and according to them the most significan effect of using HH alone was a slight increase in false detections (compared to AARI ice chart LFI).

10) P5,L26-31: A repeat of earlier in the paper.

I think it is also an important conclusion that over whole Arctic and/or Antarctic only HH mode SAR data are available from SENTINEL-1 and any operational pan-Arctic/Antarctic operational system should be based on HH alone.

11) You say how quick this algorithm runs. Why do you need to use a LFI search area mask then? Why not run it everywhere in the ocean?

It is true that the algorithm is fast for one day case. However when running it for a time series I wanted to fasten the processing by applying the mask. Actually, using the mask does not have effect on the result, it only fastens the processing. In computing longer time series the time saving is significant, for example the time to compute the results for the whole study period can be reduced by 12 hours. Also in possible future operational production fast delivery is a useful property.

12) I'm left with a lingering desire for a map of average fast ice coverage, even though your time series is very short. This would be a nice result for this paper, and allow comparison to the earlier Divine work. Somethin akin to the Fraser et al., 2012 figure 2 (https://doi.org/10.1175/JCLI-D- 10-05032.1) would be ideal.

Added/updated figures: LFI entent time series figure updated by adding three different areas of Kara Sea (as in Divine 2004), and a figure describing the relative time fraction of existence of fast ice at the grid points.

Table 1: I have no idea what the column headings represent. (A(IC), etc.) This whole table is very poorly described. Figure 1: This map is poorly presented, and appears to

[Figure]

be of low resolution. Borderline illegible.

Table 1 has been updated and simplified, I hope it is more clear and understandable now.

Figure 2: Completely illegible.

Figure 2 has been removed.

Figure 3: The choice of a black mask is not appropriate given the high amount of black in the right sub-figure. The figure needs to be made bigger. An overlay needs to be included (lat/lon, coastlines, etc.). Coordinates and north arrow have been included.

Land mask color has been changed. The coordinates and north arrow have been added.

Figure 4: Caption doesn't indicate that this is for FMI-B. Iquestion the necessity of this figure too.

The figure is actually for FMI-A, FMI-B is derived from multiple FMI-A results by the logical AND operation. FMI-A has been included in the caption.

Figure 6: The caption should refer to FMIA and FMI-B. Again, overlay needed (lat/lon/colour legend for upper left). This figure seems redundant with Fig 7 also included. Figure 7: Need a legend describing all 8 potential colours used in this figure. Also overlay, etc.

The figures have been replaced in the updated version.

Figure 8: No comment on why the AARI charts underestimate fast ice compared to your work.

It depends on the selection of parameters, in the new version the algorithm parameters have been selected such that the detected FMI-A LFI area is close to the AARI LFI area. For the new parameters this comment is not relevant.

Figure 8: Doesn't the value of 0 kmËĘ2 fast ice at day 180 in the FMI-B algorithm indicate that your coast mask is just fine? This is contrary to your statement in the text that some land pixels are not correctly masked out.

There have been some kind of error in the previous version FMI-B time series at that point, it should not go to zero. And either in the results with the new parameters of the revised manuscript it does not go to zero in the summer. The summer ice areas have now been masked off (based on FMI-A August 2016) and after this correction both LFI-A and LFI-B show zero LFI extent during the summer.

Thank You once again for Your comments! I hope the revised version has improved compared to the first version and we are iterating towards the correct direction!

Juha Karvonen, FMI

Please also note the supplement to this comment:
https://www.the-cryosphere-discuss.net/tc-2017-260/tc-2017-260-AC1-supplement.pdf

**Supplement:**

[revised manuscript text omitted]

---

## Author Comment (AC2) · 5 Mar 2018

General Comments:

The manuscript presents a new method for the landfast ice mapping based on SENTINEL-1 SAR imagery. The method is tested in the Kara and Barents Sea area and the resultant landfast ice (LFI) product compared to operational sea ice charts from Arctic and Antarctic research Institute, Russia (AARI charts). The results indicate the potential to derive reliable fast ice extent operationally. Unfortunately missing methodological details, inconsistent results and the large number of typographical and formating errors do not make an impression of a self-contained manuscript.

Thank You for the valuable comments. I have tried to improve the manuscript according

to the reviewer comments, tried to improve the description of the methodology, description of the results and manuscript layout and language. If seen necessary I can still try to improve the English language and also if seen neccessary find asuitable perosn to make an English language proofreading.

Specific comments: 1. Missing discussion of other relevant studies The introduction give an overview on of existing methods of fast ice detection, including several SAR-based methods. However, it is not clear what is the potential advantage of the proposed algorithm compared to the existing ones. p1, line 21-22: The author states: "in the fast ice zone only thermodynamic ice modeling is necessary as the modeled dynamic part can be omitted". Fast ice can be formed dynamically, it also can breakup in response to the dynamical forcing. Please, clarify your statement. It would be good to provide some references to model studies to support your statements on p 1, line 21-23.

Clear advantage compared to methods based on ice drift is that ice drift detection is very slow compared to direc cross-correlation computation. The advantage compared to cross-correlation minimum is that average and median are more robust to errors than minimum which is only one value instead of statistics. Methods based on SAR backscatter do not wotk in all conditions as SAR backscattering from sea ice varies according to many physcial parameters, for example including surface and snow cover wetness and SAR incidence angle. Have included a few sentences on this in the concluding section.

2. Missing clarity in the methodological sections In general, the description of the work-flow is confusing. It needs to be clarified in order to be reproducible: Were raster (gif, png) or vector (shp) AARI charts used? In general, the vector format is more convenient to work with. Fig 2. and Fig 6. (upper left) suggest that the raster format was used. What kind of software was used to re-project the rasters and extract fast ice extent?

I have tried to improve this part also. Also trying to make clear that the raster AARI ice

charts were used in this study.

I encountered that different product might give quite variable fast ice extent during its development in fall. It would be interesting to see whether the FMI methods show result in similar to the AARI fast ice extent in October-November. The Sentinel-1-based time series of LFI extent were derived from October 2015, however the comparison with the AARI charts covers a period from November on. What is the reason for shortening the comparison period?

I have included a monthly comparison of the classification accuracy w.r.t. AARI ice chart LFI. In August-October 2016 there was not much LFI and comparison was not feasible. A shorter time period was used in the original study made for the Horizon-2020 EC project SPICES with the data at my disposal at that time. I have now extended the AARI comparison to cover a whole year time span.

p 4, line 16-17: The author should clarify what is "the daily LFI product". Is it a product of FMI-A method? "Daily mosaics" seems to contain SAR data collected for a period of several days. The consecutive mosaic might have several days apart. In this case, the LFI product contains information from different dates. The construction of "daily mosaics" is described in 2.3 (p 3, lines 12-14). Later it becomes clear (p 3, lines 30-31) that some parts of a daily mosaic might remain from a previous day. The author should explain better how and for what time period the mosaics are constructed.

The daily product is based on the daily mosaics and naturally also the mosaics of the preceding two week time period have been used in the averaging process. Not all SAR mosaic locations (grid points) are actually updated daily, but the mosaics are continuously updated daily, so it is called a daily mosaic and daily product. Daily means that the LFI extent using the most recent mosaics and mosaic history of the preceding two weeks is issued daily. The SAR mosaic is updated on average in about two days after launching of SENTINEL-1b (late April 2016) even more often , but at some grid locations the values can be even three days old. Construction of the mosaics has been

tried to be explained in more detail in the revised manuscript.

The use of 0.19 and 0.15 TCC threshold values are not explained. In general, the values seem to be rather low. According to Fig. 6, the threshold values work well for discriminating between fast ice and open water or newly-formed ice, but appear to work worse when fast ice is surrounded by the first-year ice (area south-east of the Severnaya Zemlya Archipelago). Were the thresholds picked based on a sensitivity study?

The threshold in the previous version were defined experimentally based on visual interpretation of a few images. In the updated version the threshold have been selected based on a comparison to AARI LFI and selecting the optimal (minimizing the error) value. Also figures showing the error as a function of the threshold have been included in the updated version to shown the sensitivity. The four AARI ice charts used in defining the thresholds have been excluded from the evlauation of the results.

p 5, line 15. The temporal average and temporal median are mentioned for the first time. Please, provide more information in the methodology and results sections.

By these I mean temporal cross-correlation average and median. I hope this is more clearly explained ibn the revised mansuscript. also see my response to reviewer 1 on this topic.

p 5, lines 26-31: Application of an extended land mask would help to properly compare your results with AARI charts. First, it would exclude summer fast ice. Second, it will reduce the difference between AARI and FMI fast ice extent (in Fig. 8) and therefore add more value to the number describing the remaining differences. I suggest that the extended land mask should be applied at least for the data comparison.

Extended land mask was not applied. Instead the summer (August 2016) LFi detected by FMI-A were excluded. The same erroneous LFI areas also appeared in the summer 2017 FMI-A results and it was concluded that these are because of small errors in the

positioning of the land mask at certain locations.

3. Inconsistency of the results

p 4, lines 23-24: The description method performance does not agree with figures and table. The qualitative description "quite good" is not in line with the quantitative characteristics presented in Table 1. According to the Fig. 8, FMI methods slightly overestimate fast ice area compared to the AARI charts. If I understand the numbers in Table 1 correctly, at least half (92.+-42.3% ) of fast ice detected by FMI-A method is not actually fast ice on the AARI charts. A large area of spurious fast ice is located far from the coast between the Severnaya Zemlya and the mainland (Fig. 6, 7). Its presence can not be explained by different land mask, as suggested on p 4, line 27-28. The author claims that his methods are more precise than the AARI charts. Currently, operational charts is the most consistent and reliable source of information on fast ice extent. A cross-comparison of two data sets does not reveal a more precise method, but rather gives information about relative performance of the two. Fig. 6, 7 show that some fast ice areas (FMI-A and FMI-B) are detached from the coast, which suggests that automated methods are less precise, than the AARI charts. As the author says, different fast ice definition may indeed explain mismatch between data sets. The author however should explain what are the differences in fast ice definitions and how they may affect fast ice detection process. The given definition: "our automated algorithms locate the ice areas which has been static over a given time" is misleading. The presented method is based on reveling areas of low changes in surface backscatter characteristic, which does not ultimately mean that the ice (or any other surface) was motionless. It is also not clear what is the "given time".

The text has been updated.

LFI area from FMI-B methods differs on Fig. 8 and Fig. 9. On Fig. 8 the maximal FMI-B LFI extent is reached between Julian days 100 and 120 (April-May); it is roughly 35 000 km2. The LFI extent for the same time period on Fig. 9 (170 - 200 days from
Oct 15, 2015) exceeds 40 000 km2. Which of the figures is correct?

The numbers were incorrectly computed in the previous version, they have been up-dated to correct numbers now. Also it should be taken into account that the time step in the time series with the AARI data is one week and in the full time series it is one day. Some abrupt changes can not be seen in the weekly time series.

4. Questionable conclusions

p 5, lines 13-14: The author states that the method is suitable for operational LFI mon-itoring. Taking into account presence of large areas of spurious fast ice (Fig. 6, 7), inconsistent fast ice extent presented in Fig. 8, 9 and results from pixel-wise compar-ison with AARI charts (Table 1), I question that at this stage the methods can be use for reliable fast ice detection.

The revised results with thresholds defined based on the AARI ice chart LFI are closer to AARI ice charts. It should also be taken into account that AARI ice charts also have their error sources. also the time span of AARI ice chart input data and the FMI algorithms are differemnt (a few days vs two weeks).

5. Figures and table require a better explanation

The technical information shown in Fig. 1, 2, 5, 6, 7 can be presented more efficiently. E. g. the the study area (Fig.1) and the land mask and LFI mask (Fig. 5) can be shown in one figure. The AARI ice chart (Fig. 2) is duplicated in Fig. 6 (upper right corner). Fig. 6, 7 show the same information. Table 1 is poorly explained. Please name the steps in the flowchart (Fig. 4) in consistency with the text. E. g. Cross-corr. Is TCC in the text; area mask is referred as a mask in the text. What doest Pixelw average stand for? Please, describe in the text. All figures require better captures, legend, geographic information and land mask (if applicable).

Many of the figures have been updated. The study area figure and the mask figure are still separate figures, in my opinion int is clear this way and they can bhe placed

in a suitable place within the manuscript as separate images. after all, in an electronic publication, like TC, the number of figures is not that relevant as it may be in a traditional printed media.

The figures and figure captions have been updated according to the erviewer comments. Some figures with not much contribution have been removed/replaced. Also the text referring to the figures has been updated.

Technical corrections: p 1, line 2: Please replace "ove Kara and Barents Seas" by "over the Kara and Barents Seas"

Corrected.

p 1, line 8: Please remove excessive spaces before commas in citations here and throughout the text

I think these are automatically generated by latex based on the copernicus template.

p 1, line 11: Missing citation after Zubov, 1945

Corrected.

p 1, line 12-13: Do Yu et al. (2014) indeed give this number in their paper? Please, rephrase, in case the 13% is not mentioned by Yu et al. (2014).

It is not directly said in the reference, but can be derived from the numbers given. I have tried to rephrase this.

p 1, line 13: Please replace "sea ice coverage" by "sea ice cover"

Corrected.

p1, line 16: "quite similar criteria" is kind of vague. Please clarify.

Some explaining text has been added.

p 1, line 20: Wrong citation. To support your statement, use the work by Maqueda, M.,

[Figure]

Willmott, A.J. and Biggs, N.R.T., 2004. Polynya dynamics: a review of observations and modeling. Reviews of Geophysics, 42(1). The importance of fast ice was not studied by Selyuzhenok et al. (2015). The paper rather describes changes in the fast ice regime. Please, move the reference to p 1, line 18 : "later formation and earlier disappearance (Mahoney et al. , 2014, Selyuzhenok et al., 2015)".

Thank You for the reference! cjanged as suggested and reference added.

p 1, line 24-25: "The proposed method has been used and will be used for: : :" Has the method been used before? The sentence seems to be out of the context. Please move it to the end of the introduction, where the proposed method is introduced.

The method has been used in this study and will be used in completing the time series (next time will likely be in spring 2018 after the busiest Baltic Sea ice season will be over).

p1, line 25: What is the existing LFI time series? Are you referring to the AARI charts or another product? Please clarify.

At least there are ice charts, including AARi ice charts. I am not aware of all possible time series, not all of them are public. Added ice charts in the sentence.

p 2, line 8: Please replace " in the case on" by " in the case of"

This is not included in the revised text any more.

p 2 line 14-15: The sentence starting with "In Mahoney et al. 2004, 2005.." sounds as the fast ice was detected based on mosaic edge, orientation and temporal difference. I suggest changing to "In Mahoney et al. (2004, 2005) LFI is detected based on vector grayscale gradient fields of 3 subsequent SAR images"

Changed as suggested.

p 2, line 2: Replace "re-reprojected" by "reprojected"

[Figure]

Changed.

p 3, line 21: adjacent daily SAR mosaic?

Changed.

p 3, line 23: in Fig 4.?

Changed.

p 3, line 25: To increase computing performance and to exclude: : : ?

Changed.

P 3, line 29: Please replace "i,e, white areas in Fig. 4" by "i. e. white areas in Fig. 5"

In the revised version it is Fig. 4.

p 4, line 1: " less than zero", Did you mean "less than one" or it is indeed negative?

Yes, changed.

P 4, line 8: Please remove "still" in " We still additionally applied.." p 4, line 12: Please remove "still" in " we still additionally perform.."

Text has been changed.

p 4, line 12: Please replace "logical and operation" by "logical AND operation"

AND now written with capital letters.

p 4, line 13: Please remove "in this context"

REmoved.

p 4, line 15: Typo in "results" p 4, line 19-22: Inconsistent terminology: FMI algorithms, SAR algorithms p 4, line 25: Missing figure number (7) p 4, line 32-33: Duplicated "whole study" and "our study area" p 5, line 3: Typo in "erroneous" p 5, line 10: Typo in "developed"

These have (hopefully) disappeared in the revised text. Much of the text has been updated in the revised version.

I hope the revised version has improved compared to the first version and we are iterating towards the correct direction!

Thank You for Your comments!

Juha Karvonen, FMI

Please also note the supplement to this comment:
https://www.the-cryosphere-discuss.net/tc-2017-260/tc-2017-260-AC2-supplement.pdf

**Supplement:**

[revised manuscript text omitted]
 Conditions and Hazards and Chukchi and Bering Seas, Report of Conditions and Hazards and Chukchi and Bering Seas, Report of Conditions and Hazards and Chukchi and Bering Seas,
- 35 Geophysical Institute, University of Alaska, Fairbanks, AK, USA, 130 pages, 1980.
  Sun, Y., Automatic ice motion retrieval from ERS-1 SAR images using the optical flow method, Int. J. Remote Sens.International Journal of
  - Remote Sensing, v. 17, n. 11, pp. 2059–2087, 1996.

Thomas – M., Geiger, C. A., and Kambhamettu, C., High resolution (400 m) motion characterization of sea ice using ERS-1 SAR imagery, Cold Regions Science and Technology, v. 52, n. 2, pp. 207–223, 2008.

Weeks - W. F., On Sea Ice, University of Alaska Press, Fairbanks, Alaska, USA, ISBN 978-1-60223-101-6, 2010.

5

- Wessel , P., W. H. F. Smith, A Global Self-consistent, Hierarchical, High-resolution Shoreline Database, J. Geophys. Res. Journal of Geophysical Research, v. 101, n. B4, pp. 8741-8743, DOI: 10.1029/96JB00104, 1996.
- WMO, WMO Sea-Ice Nomenclature, World Meteorological Organization, Report No.259, available online: http://www.jcomm.info/index.php?option=com\_oe&task=viewDocumentRecord&docID=14598, 2015.
  - Yu, Y. L., H. Stern, C. Fowler, F. Fetterer, J. Maslanik, Interannual variability of Arctic landfast ice between 1976 and 2007, J. Clim., v. 27, n. 1, pp. 227–243, doi:10.1175/JCLI-D-13-00178.1, 2014.
- 10 Zubov, N. N. (1945), Arctic Sea Ice(in Russian), Izd.Glavsevmorputi, Moseow, 1945. Arctic Ice, translated from the Russian original "Ledy Arktiki" (Moscow, 1945), U.S. Navy Electronics Laboratory, 1960.

Table 1. Comparison by of the FMI methods to AARI ice charts, the numbers are in percents. The values in parentheses are standard deviations in percentage points of the AARI ice chart LFI.

| Method                            | A(ICDetected (%)                                      | A(SAR IC)A(IC ~SAR) A(SAR ~IC)False detection (%)             |
|-----------------------------------|-------------------------------------------------------|---------------------------------------------------------------|
| FMI-A <del>(km2)</del> | <del>33969 27719 6250 31387(%73.1 (8.8</del> ) | <del>100 81.6 (17.2)18.4 (17.2) 92.4 (42.3) 20.9 (11.8)</del> |
| FMI-B <del>(km2)</del> | <del>33969 23303 10666 12738(%50.4 (13.2</del> )      | <del>100 68.6 (21.9)31.4 (21.9) 37.5 (11.4)4.3 (2.2)</del>    |

**Figure Captions**

5

Figure 1. The study area in the used polar stereographic projection.

Figure 2. AARI ice chart of March 8, 2016. ©AARI.

Figure 3. SAR mosaics of March 8, 2016, HH mosaic (left panela) and HV mosaic (right panelb). The land areas , based on our land mask, appear as green and areas of no data are as black in the mosaics figures.

Figure 4. 3. Block diagram of the LFI detection (FMI-A) for SENTINEL-1 HH polarization channel. The process for the HV channel is similar, except a threshold value of  $T_{HV}$  is applied instead of  $T_{HH}$ .

Figure 5.4. Mask used to locate the areas where LFI is searched. White areas are the indicate the LFI search area, gray green areas are land.

10 Figure 5. The total number of erroneously classified pixels as a function of the temporal cross-correlation average for HH channel SAR data (a) and for HV channel SAR data (b). The optimal thresholds were defined as the minimum of the total error ("sum" curves according to the figures legend).

Figure 6. ARI AARI ice chart of March 8, 2016, translated to the polar stereographic projection used in this study , LFI based on the and cropped to the study area.

15 Figure 7. LFI extent based on AARI ice chart (black areas), LFI based on the FMI method, and LFI based on the FMI method with additional temporal filtering.

Figure 7. A false color image a), FMI-A LFI (b) and FMI-B LFI (c) of March 8, 2016, combining the products in comparison. In the False color image Red=AARI 2016. LFI areas are the black areas in the figures.

Figure 8. Monthly detection and false detection percentages for LFI-A (a) and LFI-B (b) compared to AARI ice chart LFI,

20 and the relative amount of (AARI) LFI points (c) in percents of the LFI , Green=FMI-A LFI and Blue=FMI-B LFI . In the white areas all the three LFI estimates agreepoints of the whole year.

Figure 8. Time series of the weekly (from Nov 3.9. Ice extent time series of AARI ice charts, FMI-A and FMI-B during the one-year period from November 1 2015 until July 5 2016) LFI extent over the study according to FMI-A, FMI-B and October 31, 2016. The time series is weekly with FMI-A and FMI-B for the same days as the weekly AARI ice charts.

Figure 9. A LFI area times series of 10. FMI-A (a) and FMI-B (b) LFI time series for the whole study period from October 15 2015 to August 30 2017, using the until August 31 2017. Also the time series of Kara Sea sub-regions, southwestern (SW), northeastern (NE) and Gulf of Ob (Ob) have been included in the figures.

Figure 11. Temporal LFI coverage (percentage) during the period from November 1 2015 until October 31 2016 based on the weekly AARI ice charts (a), daily FMI-A (b) and daily FMI-B algorithm(c).

---

## Author Comment (AC3) · 18 May 2018

Well done, Dr Karvonen, for addressing my original comments very nicely. Aside from very minor English comments which will hopefully be addressed in the proof stage, I suggest you take a closer look an new Figure 11. Some of the text is half-missing, and the coloured legend at the bottom needs some kind of text to indicate what it shows.
* * *
Thank You for the comments.

I have tried to check the languaga and also performed spell checking for the updated manuscript.

[Figure]

I have also updated figure 11. The subfigures were put too tightly by latex and I changed the latex figure paramters (vspace). The colorbar indicates the percentage of time with fast ice during the whole year. An explanation has been included in the capture and the colorbar has been updated to include the percent character.

Thank You! Juha Karvonen, FMI

---

## Author Comment (AC4) · 18 May 2018

Review of Estimation of Arctic Land-Fast Ice Cover based on SENTINEL-1 SAR Imagery, Juha Karvonen by Valeria Selyuzhenok General comments: Overall, most of my comments were considered. The manuscript has improved after the revision. The methodology is presented in a more clear way and the new figures allows for easier evaluation of the methods performance. This new information triggered additional questions. Please, consider the following comments:

Thank You for the comments, in the updated version and this reposnse I have tried to address all Your comments. i hope the clarity of the manuscript has improved from the previous version.
* * *
1. The introduction presents a short overview of landfast ice studies including different methods of fast ice detection. It is not clear why a new method is required and how it would contribute to scientific progress. I suggest to add few sentences clearly describing the objective of this study. Page 2, lines: 31-33: "The algorithms proposed in this study are used for creating daily time series of the Kara and Barents Sea LFI extent in high-resolution gradually complementing the existing Arctic LFI time series derivable from Arctic operational ice charts." It should be clear to the reader why it is important to produce daily (not bi-weekly or weekly) data set and in what sense the new data set would complement operational charts?

The daily products are produced because also other FMI sea ice products (sea ice thickness and concentration) over the are are daily (operational tests have been made during a few winters and they are continuously run daily) and the spatial resolution also better corresponds to our products. And we have plans to utilize the LFI estimation in our ice thickness estimation, thus a daily product in high-resolution is useful for us. For more detais, see my responses later.
* * *
Doesn't actually FMI-B produce 2-weeks average fast ice product?

Not exactly, FMI-B provides the areas which have been classified as LFI in each day of the two week period. FMI-B can be used to identify the areas which very likely represent LFI. Parts of LFI areas will be missed by FMI-B, however. It looks that in most cases the LFI detected by FMI-B are the same areas but the areas are smaller than for AARI ice charts or FMI-A. Thus FMI-A is preferred to be used for daily LFI detection. However, FMI-B could be used if we want to locate the only static (LFI) with a very high confidence and not include areas which have a bit lower (but still high) likelihood of being LFI (according to the automated algorithms).
* * *
2. The methodological section has improved, but I find it a bit difficult to follow. I suggest restructuring the section in the way that it consecutively describes each step of the algorithm as they indicated in Figure 3. Please, avoid duplication: the paragraph on page 5 starting from line 10 seems to be an extended version of the text on page 4, lines 17-28.

I have updated the description and included references to Fig. 3 in the text. Also tried to remove repetition.
* * *
3. Discussion and conclusions require major modification - Misleading interpretation of the results Without knowing the purpose of the study, it is difficult to judge whether the performance of the developed methods is good enough. For some purposes it might be important to know that probability of fast ice presence in the detected area is very high. Than, indeed FMI-B method would be more reliable, compared to FMI-A. However, FMI-B product would be irrelevant to quantify changes in fast ice cover, since it detects only half of fast ice area.

We at FMI think that FMI-A algorithm results are good enough to include the LFI detection in the operational FMI Arctic and Baltic products and to use the LFI also in improving the ice thickness estimation in these areas. This is true especially during the winter months (approx November-April). FMI-B was included here just as a reference, it may have applications in combining the two methods in an optimal way in the future (depending on funding and other future resources, unfortunately the EC HORIZON-2020 SPICES project funding used for this study has already been used and in that framework it is not possible to perform additional devleopment) The purpose of the study is also related to the automated sea ice products based on combined ice modelling and data from multiple EO input data sets, for more details, see my responses later.
* * *
Page 8, line 21: "FMI-B can then be considered as an algorithm locating the LFI areas with a high confidence."

Have changed this sentence to indicate that "FMI-B can be considered as an algorithm locating only the areas which very likely represent LFI." I hope it is more clear this way.
* * *
FMI-B algorithm systematically underestimates the LFI area (Fig. 9, Tab.1). It shows only 50% of fast ice presented on operational charts. The author also mentions that fast ice edge location is not presented correctly compared to AARI charts (page 6, lines 28-30). I assume that at this stage, the algorithms can not be considered as a reliable method to map fast ice operationally. I recommend that further improvements (e.g. suggested by the author on page 10, lines 6-11) are made in order to provide more reliable operational data.

Unfortunately FMI does not have resources to make this kind of algorithm development studies/work now as the SPICEs funding for the related WP has ended. We think the results are useful and we are going to apply FMI-A in our operational tests and studies. Our aim is to include LFI detection in our operational framework both in the Baltic Sea and in the Arctic study area in the first phase. Wea re curently estimating ice thickness in Baltic and in the Arctic study area. In the Arctic study area we use TOPAZ-4 model as a background information, but the TOPAZ data has not proved to be very reliable and in the future versions we are going to include radar altimeter data (Cryosat-2) and use the FMI thermodynamic model HIGHTSI in the static ice areas during the static periods to estimate the themodynamic growth/melt in these ice fields to get good estimates of ice thickness. This has already been tested over the Baltic Sea LFI during the winter 2017-2018. We have studied the use of HIGHTSI model and SAR data in ice thickness estimation in multiple enclosed or semi-enclosed sea areas (such as Baltic Sea, Caspian Sea, Gulf of St. Lawrene and Bohai Sea) with promising results.

However, in a non-enclosed seas our aim is to apply HIGHTSI only over static ice areas (during their being static). The LFI will be produced daily in the similar manner as ice thickness and ice concentration (based on SAR and microwave radiometer). A new algorithm was needed to get LFI in the same spatial and temporal scale as the other sea ice products. It is true that these do not always give exactly the correct ice information, however, we at FMI think they would be useful in navigation and also could be used to assist the visual/manual ice analysis.

I have added some text (and references) on this in the introduction.

References related to combining SAr imagery and an ice model for ice thickness estimation:

Arctic sea ice thickness (SIT): Markku Simila, M. Makynen, J. Karvonen, A. Gegiuc, A. Gierisch MODELED SEA ICE THICKNESS ENHANCED BY REMOTE SENSING DATA May 2016, Conference: Living Planet Symposium 2016

Gulf of St. Lawrence SIT: J. Karvonen, B. Cheng, T. Vihma, M. Arkett, and T. Carrieres, A method for sea ice thickness and concentration analysis based on SAR data and a thermodynamic model, The Cryosphere, v. 6, pp. 1507-1526 (http://www.the-cryosphere.net/6/1507/2012/tc-6-1507-2012.html), 2012.

Caspian Sea SIT: Karvonen, J.; Cheng, B.; Vihma, T. Estimation of Sea Ice Parameters Based on X-Band SAR Data and Thermodynamic Snow/Ice Modelling for the Caspian Sea. In Proceedings of the International Conferences on Port and Ocean Engineering under Arctic Conditions (POAC'13), Espoo, Finland, 9–13 June 2013; Available online: http://www.poac.com/Papers/2013/pdf/POAC13_029.pdf (accessed on 3 May 2018).

Balti Sea SIT: J. Karvonen B. Cheng, M. Simila, Ice Thickness Charts Produced by C-Band SAR Imagery and HIGHTSI Thermodynamic Ice Model, Proc. of the Sixth Workshop on Baltic Sea Ice Climate, pp. 71-81, Lammi, Finland 2008.

Bohai Sea SIT: Juha Karvonen, Lijian Shi, Bin Cheng, Markku Similä, Marko

Mäkynen, Timo Vihma, Bohai Sea Ice Parameter Estimation Based on Thermodynamic Ice Model and Earth Observation Data, Remote Sens. 2017, 9(3), 234; https://doi.org/10.3390/rs9030234 https://www.mdpi.com/2072-4292/9/3/234/pdf (accessed on 3 May 2018).

It should also be taken into account that very also the AARI ice chart LFI includes some kind of error sources and inaccuracies. One definitely comes from the fact that the whole Arctic is a large area and an ice analyst is not able to make as accurate (e.g. 500m pixel size) analysis as a computer over such a larger area and the drawn polygons tend to smooth the boundary lines of different ice fields. It is also a fact that diffrent ice analysts see the ice situation in their own ways and there are differences between the ice analyses depending on the ice analyst (this was shown for sea ice concentration in the paper based on an experiment made in the IAW-2014 workshop). An algorithm provides systematical estimates and eliminates the variability due to possible varying interpretations and skills of different ice analysts.

Also added some text on this.

Reference with some information on the differences in analyzing the same ice field sea ice concentration (SIC) by separate ice analyst groups (not even individuals, probably producing even more deviating SIC information:

A comparison of SIC estimates provided by different ice analyst groups: Juha Karvonen, Jouni Vainio, Marika Marnela, Patrick Eriksson, Tuomas Niskanen A Comparison Between High-Resolution EO-Based and Ice Analyst-Assigned Sea Ice Concentrations, IEEE Journal of Selected Topics in Applied Earth Observations and Remote Sensing, v. 8, n. 4, pp. 1799 - 1807, 2015 DOI: 10.1109/JSTARS.2015.2426414

Included this here just to indicate that different ica analyst produce different results (in the publication for SIC but very likely also for polygon boundaries and the other ice properties within them).
* * *
- Missing references

Page 9, lines 12-22: Comparing the presented method with other studies, the author does not provide any references. Please, refer to literature to support your conclusions.

Some references are already given in the intro section. I added some references in this section to indicate which specific methods were considered here.
* * *
- Redundant information Page 7, line 13– page 8, line 2: The comparison of air temperature measurement from the Longyearbyen weather station with variations in annual fast ice development neither belong to this methodological paper, nor present relevant scientific results. Linking 2 years of fast ice annual cycle with air temperatures measured 1500 km away from the study area does not make sense to me. First, there are several studies investigating fast ice development in the Kara Sea (Divine et al. 2003, 2004, 2005; Olason 2016) which indicate that air temperature is not the only factor controlling fast ice cover. Second, data from the Longyearbyen weather station are not representative for the study area because it can be affected by different atmospheric circulation regimes. To use such data, the authors should first prove that at least the atmospheric circulation over Svalbard and the Kara Sea were similar during the season. It might be more reliable to use reanalysis data. I recommend removing these paragraphs.

I have updated and shortened these paragraphs. I also studied the NCEP/NCAR reanalysis air temperature data at two locations, one in the southern Kara Sea and another in the northern Kara Sea. The temperature time series during the study period in the northern Kara Sea were quite similar to those of Longyearbyen, in the south some warmer during the summer, but the mostly including the same (warmer and colder) periods as for the two more northern locations. These all three indicate that the win-

ter 2016-2017 was more severe than winter 2015-2016, and also the LFI extent was larger for the more severe winter of 2016-2017. It should also be noted that the stydy area is not only limited to Kara Sea but also the Svalbard area (and Longyearbyen) were included in the study area. Lonyearbyen was selected because the data were available. Unfortunately, we do not at FMI have access to Russian measurements. It would be nice to discuss with the Russian institutes of future research co-operation also including data sharing with each other. I think it would be a benefit for both.

—————————————————————————————————————————

4. In general the text is difficult to read. I feel that the English may need improvements.

I have tried to check and further improve the language. The other reviewer was a native English speaker and he was able to understand my English. There may be some details needing correction, but I hope this will be possible in the final editing phase (if the manuscript will be published). If found necessary by the TC editor, a proofreading service can also be used for the final version.

—————————————————————————————————————————

Specific comments:

Page 1, lines 23-24 – page 2, lines 1-2 : The part regarding fast ice modeling was marked for improvement during the interactive discussion. It has became even more confusing. It is not clear why author starts talking about fast ice thermodynamic and dynamic modeling. The paragraph does not seem to bare any relevant information. I suggest clarifying the message and including references to modeling studies or removing the entire paragraph.

I have tried to clarify the context. We have plans to apply a thermodynamic ice model over the static ice fields to estimate ice thickness. We have done this over the Baltic Sea succesfully during this year's winter. We'll continue the Baltic work next winter and will provide an operational Baltic LFI service demontration (a CMEMS downstream service BALFI). After the season 2018-2019 we'll also be able to provide some accuracy comparisons over the Baltic, but it will be a topic of another publication which will also involve a detailed description of the service demontration still partly under construction.
* * *
Page 6, lines 7-8: The FMI-B underestimates LFI, compared to AARI charts (Table 1, Figure 9). It doest not seem to perform better, than FMI-A. Although FMI-B produces less false positive estimates, the number of true positive is also reduced. Would FMI-A produce similar results applying higher threshold?

It is true that FMI-B produces significant underestimation of LFI. It can possibly be used to locate the areas of LFI which very likely represent LFI. Possibly it could be then combined with LFI-A to gro the LFI-B LFI areas e.g. by those areas of LFI-A which are connected to the LFI-B areas (region growing). This will be a topic of future research depending on resources available. currently no resources for further Arctic LFI studies exist, the only thing we can do is torun and update the time LFI series ob ftp.
* * *
Figure 8 : It is not clear to me what is shown in (c).

It gives the relative amount of LFI in AARI ice charts during each month of the year. This is to help to interpret the other subfigures, e.g. in June there were about 5% of the total LFI and in July only about 2% of the total LFI grid points, so even though there were more erroneous classifications compared to AARI ice charts during these months, their contribution to the total error was small compared to winter months with over 15% of the total LFI grid points. These are simply computed as (monthly) percentages of the sum of the whole LFI areas of the one year period AARI ice chart LFI. I have added an explanation in the caption.

Also corrected an error in the y-axis labels, there should naturally be "5" and "10"

instead of "0" and "5", this may also have caused problems with interpretation of the figure.

———————————————————————————————————————

Figure 10 : What are the boarders of the regions (SW, NW, Ob)? Why do these regions come into play? It would be more useful to provide the curve for the entire area to compare it with the AARI data.

A sub-figure indicating the area division used has been added. This is just to enable comparison to the results of the given reference using this kind of division. Unfortunately, inclusing more AARI data would require much work with the data and we do not have the resources for this work now (the supporting project SPICES is ending and no more work related to this WP of this project can be made any more).

———————————————————————————————————————

Technical comments: The text requires careful proofreading to exclude typographical errors

I have perfomed spell checking and tried to improve the language.

———————————————————————————————————————

Thank You! Juha Karvonen, FMI

---

## Referee Report (RR1)

**Review of Estimation of Arctic Land-Fast Ice Cover based on SENTINEL-1 SAR Imagery, Juha Karvonen**

by Valeria Selyuzhenok

General comments:

Overall, most of my comments were considered. The manuscript has improved after the revision. The methodology is presented in a more clear way and the new figures allows for easier evaluation of the methods performance. This new information triggered additional questions. Please, consider the following comments:

**1.** The introduction presents a short overview of landfast ice studies including different methods of fast ice detection. It is not clear why a new method is required and how it would contribute to scientific progress. I suggest to add few sentences clearly describing the objective of this study.

Page 2, lines: 31-33: "The algorithms proposed in this study are used for creating daily time series of the Kara and Barents Sea LFI extent in high-resolution gradually complementing the existing Arctic LFI time series derivable from Arctic operational ice charts."

It should be clear to the reader why it is important to produce daily (not bi-weekly or weekly) data set and in what sense the new data set would complement operational charts?

Doesn't actually FMI-B produce 2-weeks average fast ice product?

**2.** The methodological section has improved, but I find it a bit difficult to follow.
I suggest restructuring the section in the way that it consecutively describes each step of the algorithm as they indicated in Figure 3. Please, avoid duplication: the paragraph on page 5 starting from line 10 seems to be an extended version of the text on page 4, lines 17-28.

**3.** Discussion and conclusions require major modification

 - Misleading interpretation of the results

Without knowing the purpose of the study, it is difficult to judge whether the performance of the developed methods is good enough. For some purposes it might be important to know that probability of fast ice presence in the detected area is very high. Than, indeed FMI-B method would be more reliable, compared to FMI-A. However, FMI-B product would be irrelevant to quantify changes in fast ice cover, since it detects only half of fast ice area.

Page 8, line 21: "FMI-B can then be considered as an algorithm locating the LFI areas with a high confidence."

FMI-B algorithm systematically underestimates the LFI area (Fig. 9, Tab.1). It shows only 50% of fast ice presented on operational charts. The author also mentions that fast ice edge location is not presented correctly compared to AARI charts (page 6, lines 28-30). I assume that at this stage, the algorithms can not be considered as a reliable method to map fast ice operationally. I recommend that further improvements (e.g. suggested by the author on page 10, lines 6-11) are made in order to provide more reliable operational data.

- Missing references

Page 9, lines 12-22:
Comparing the presented method with other studies, the author does not provide any references. Please, refer to literature to support your conclusions.

- Redundant information

Page 7, line 13– page 8, line 2:

The comparison of air temperature measurement from the Longyearbyen weather station with variations in annual fast ice development neither belong to this methodological paper, nor present relevant scientific results.

Linking 2 years of fast ice annual cycle with air  temperatures measured 1500 km away from the study area does not make sense to me.  First, there are several studies investigating fast ice development in the Kara Sea (Divine et al. 2003, 2004, 2005; Olason 2016) which indicate that air temperature is not the only factor controlling fast ice cover. Second,  data from the Longyearbyen weather station are not representative for the study area because it can be affected by different atmospheric circulation regimes. To use such data, the authors should first prove that at least the atmospheric circulation over Svalbard and the Kara Sea were similar during the season. It might be more reliable to use reanalysis data. I recommend removing these paragraphs.

**4.** In general the text is difficult to read.  I feel that the English may need improvements.

Specific comments:

Page 1, lines 23-24 – page 2, lines 1-2 :
The part regarding fast ice modeling was marked for improvement during the interactive discussion. It has became even more confusing.  It is not clear why author starts talking about fast ice thermodynamic and dynamic modeling.  The paragraph does not seem to bare any relevant information. I suggest clarifying the message and including references to modeling studies or removing the entire paragraph.

Page 6, lines 7-8:
The FMI-B underestimates LFI, compared to AARI charts (Table 1, Figure 9).  It doest not seem to perform better, than FMI-A. Although FMI-B produces less false positive estimates, the number of true positive is also reduced. Would FMI-A produce similar results applying higher threshold?

Figure 8 :  It is not clear to me what is shown in (c).
Figure 10 : What are the boarders of the regions (SW, NW, Ob)? Why do these regions come into play? It would be more useful to provide the curve for the entire area to compare it with the AARI data.

Technical comments:

The text requires careful proofreading to exclude typographical errors.

---

## Author Response (AR2)

**Estimation of Arctic Land-Fast Ice Cover based on Dual-Polarized SENTINEL-1 SAR Imagery**

Juha Karvonen[1]

[1]Finnish Meteorological Institute, PB 503, FI-00101, Helsinki, Finland

*Correspondence to:* Juha Karvonen (juha.karvonen@fmi.fi)

Dear members of TC editorial board,

I have now carefully revised my manuscript (made am major revision) and separately submitted reposnses for the reviewer comments. The updated manuscript has also bee submitted and additionally a version with the changes indicated (by latexxdiff) has been uploaded.

Sincerely, Juha Karvonen, FMI

[revised manuscript text omitted]

---

## Author Response (AR3)

**Estimation of Arctic Land-Fast Ice Cover based on Dual-Polarized SENTINEL-1 SAR Imagery**

Juha Karvonen[1]

[1]Finnish Meteorological Institute, PB 503, FI-00101, Helsinki, Finland

**Correspondence:** Juha Karvonen (juha.karvonen@fmi.fi)

Dear Prof. Haas and members of TC editorial board,

I have now revised my manuscript making some minor corrections to clarify and improve the text and extended the abstract according to the reviewer suggestion. I also included a version (latex/tex) with the table and figures placed in the included supplement zip package. Also all the related tex-files, figures and the pdf with changes tracked are included in the zip package

5    Thank You!

Sincerely, Juha Karvonen, FMI